**Developing a monthly radiative kernel for surface albedo change from satellite**
**climatologies of Earth's shortwave radiation budget: CACK v1.0**
Ryan M. Bright[1*] and Thomas L. O'Halloran[2,3]
1 – Norwegian Institute of Bioeconomy Research, Ås, Norway
2 – Department of Forestry and Environmental Conservation, Clemson University, Clemson,
South Carolina, USA.
3 – Baruch Institute of Coastal Ecology and Forest Science, Clemson University,
Georgetown, South Carolina, USA
[*]Contact: ryan.bright@nibio.no
**Abstract**
Due to the potential for land use / land cover change (LULCC) to alter surface albedo, there is
need within the LULCC science community for simple and transparent tools for predicting
radiative forcings ($\Delta F$) from surface albedo changes ($\Delta \alpha_s$). To that end, the radiative kernel
technique – developed by the climate modeling community to diagnose internal feedbacks
within general circulation models (GCMs) – has been adopted by the LULCC science
community as a tool to perform offline $\Delta F$ calculations for $\Delta \alpha_s$. However, the codes and
data behind the GCM kernels are not readily transparent, and the climatologies of the
atmospheric state variables used to derive them vary widely both in time period and duration.
Observation-based kernels offer an attractive alternative to GCM-based kernels and could be
updated annually at relatively low costs. Here, we present a radiative kernel for surface
albedo change founded on a novel, simplified parameterization of shortwave radiative transfer
driven with inputs from the Clouds and the Earth's Radiant Energy System (CERES) Energy
Balance and Filled (EBAF) products. When constructed on a 16-year climatology (2001-
2016), we find that the CERES-based albedo change kernel – or CACK – agrees remarkably
well with the mean kernel of four GCMs (rRMSE = 14%). When the novel parameterization
underlying CACK is applied to emulate two of the GCM kernels using their own boundary
fluxes as input, we find even greater agreement (mean rRMSE = 7.4%), suggesting that this
simple and transparent parameterization represents a credible candidate for a satellite-based
alternative to GCM kernels. We document and compute the various sources of uncertainty
underlying CACK and include them as part of a more extensive dataset (CACK v1.0) while
providing examples showcasing its application.
**Keywords:** GCM, radiative forcing, land use change, land cover change, LULCC
**1. Introduction**
Diagnosing changes to the shortwave radiation balance at the top-of-the-atmosphere (TOA)
resulting from changes to albedo at the surface ($\Delta\alpha_s$) is an important step in predicting
climate change. However, outside the climate science community, many researchers do not
have the tools to convert $\Delta\alpha$ to the climate-relevant $\Delta F$ measure (Bright, 2015; Jones et al.,
2015), which requires a detailed representation of the atmospheric constituents that absorb or
scatter solar radiation (e.g. cloud, aerosols, and gases) and a sophisticated radiative transfer
code. For single points in space or for small regions, these calculations are typically
performed offline – meaning without feedbacks to the atmosphere (e.g., (Randerson et al.,
2006))). Large-scale investigations (e.g. Amazonian or pan-boreal LULCC (Bonan et al.,
1992; Dickinson and Henderson-Sellers, 1988)) typically prescribe the land surface layer in a
GCM with initial and perturbed states, allowing the radiative transfer code to interact with the
rest of the model. While this has the benefit of allowing interaction and feedbacks between
surface albedo and scattering or absorbing components of the model, such an approach is

computationally expensive and thereby restricts the number of LULCC scenarios that can be

investigated (Atwood et al., 2016). Consequently, this method does not meet the needs of

some modern LULCC studies which may require millions of individual land cover transitions

to be evaluated cost effectively (Ghimire et al., 2014; Lutz and Howarth, 2015).

Within the LULCC science community, two methods have primarily met the need for

efficient $\Delta F$ calculations from $\Delta\alpha_s$: simplified parameterizations of atmospheric transfer of

shortwave radiation (Bozzi et al., 2015; Bright and Kvalevåg, 2013; Caiazzo et al., 2014;

Carrer et al., 2018; Cherubini et al., 2012; Muñoz et al., 2010), and radiative kernels (Ghimire

et al., 2014; O'Halloran et al., 2012; Vanderhoof et al., 2013) derived from sophisticated

radiative transfer schemes embedded in GCMs (Block and Mauritsen, 2014; Pendergrass et

al., 2018; Shell et al., 2008; Soden et al., 2008). Simplified parameterizations of the LULCC

science community have not been evaluated comprehensively in space and time. Bright &

Kvalevåg (2013) evaluated the shortwave $\Delta F$ parameterization of Cherubini *et al.* (2012)

when applied at several globally distributed sites on land, finding inconsistencies in

performance at individual sites despite good overall cross-site performance. Radiative kernels

(Block and Mauritsen, 2014; Pendergrass et al., 2018; Shell et al., 2008; Soden et al., 2008) –

while being based on state-of-the-art models of radiative transfer – have the downside of

being model-dependent and not readily transparent. While the radiative transfer codes behind

them are well-documented, the scattering components (i.e. aerosols, gases, and clouds)

affecting transmission have many simplifying parameterizations, vary widely across models,

and may contain significant biases (Dolinar et al., 2015; Wang and Su, 2013). An additional

downside is that the atmospheric state climatologies used to compute the GCM kernels vary

widely in their time periods (i.e., from pre-industrial to the year 2007) and durations (from 1

to 1,000 yrs). The application of a state-dependent GCM kernel that is outdated may be

undesirable in regions undergoing rapid changes in cloud cover or aerosol optical depth, such

as in the northwest United States (Free and Sun, 2014) and in southern and eastern Asia
(Srivastava, 2017; Zhao et al., 2018), respectively.  An albedo change kernel based on Earth-
orbiting satellite products could be updated annually to capture changes in atmospheric state
at relatively low costs.
The NASA Clouds and the Earth's Radiant Energy System (CERES) Energy Balance and
Filled (EBAF) products (CERES Science Team, 2018a, b), which are  based largely on
satellite optical remote sensing, provide the monthly mean boundary fluxes and other
atmospheric state information (e.g., cloud area fraction, cloud optical depth) that could be
used to develop a more empirically-based alternative to the GCM-based kernels.  The latest
EBAF-TOA Ed4.0 (version 4.0) products have many improvements with respect to the
previous version (version 2.8, Loeb et al. 2009), including the use of advanced and more
consistent input data, retrieval of cloud properties, and instrument calibration (Kato et al.,
2018; Loeb et al., 2017).
Here, we present an albedo change kernel based on the CERES EBAF v4 products – or
CACK.  Underlying CACK is a simplified model of shortwave radiative transfer through a
one-layer atmosphere.  The model form (or parameterization) is selected after a two-stage
performance evaluation of six model candidates:  two analytical, one semi-empirical, and
three empirical.  An initial performance screening is implemented where all six model
candidates are driven with a 16-year climatology (January 2001 – December 2016) of
monthly all-sky boundary fluxes from CERES, with the resulting kernels benchmarked both
qualitatively and quantitatively against the mean of four GCM-based kernels (Block and
Mauritsen, 2014; Pendergrass et al., 2018; Shell et al., 2008; Soden et al., 2008).  Top model
candidates from the initial performance screening are then subjected to an additional
performance evaluation where they are applied to emulate two GCM kernels using their own
boundary fluxes as input, which eliminates possible biases related to differences in the GCM
representation of clouds or other atmosphere state variables.
We start in Section 2 by providing a brief overview of existing approaches applied in LULCC
climate studies for estimating $\Delta F$ from $\Delta \alpha$. We then present the six model candidates in
Section 3. Section 4 describes the model evaluation and uncertainty quantification methods,
in addition to two application examples. Results are presented in Section 5, while Section 6
discusses the merits and uncertainties of a CERES-based kernel relative to GCM-based
kernels.
**2 Review of existing approaches**
Earth's energy balance (at TOA) in an equilibrium state can be written:
$$0 = F = LW_\uparrow^{TOA} - (SW_\downarrow^{TOA} - SW_\uparrow^{TOA}) \tag{1}$$
where the equilibrium flux $F$ is a balance between the net solar energy inputs ( $SW_\downarrow^{TOA} - SW_\uparrow^{TOA}$
) and thermal energy output ( $LW_\uparrow^{TOA}$ ). Perturbing this balance results in a radiative forcing
$\Delta F$, while perturbing the shortwave component is referred to as a shortwave radiative forcing
and may be written as:
$$\Delta F = \Delta(SW_\downarrow^{TOA} - SW_\uparrow^{TOA}) = \Delta SW_\downarrow^{TOA}\left(1 - \frac{SW_\uparrow^{TOA}}{SW_\downarrow^{TOA}}\right) - SW_\downarrow^{TOA}\left(\Delta\frac{SW_\uparrow^{TOA}}{SW_\downarrow^{TOA}}\right) \tag{2}$$
where the shortwave radiative forcing results either from changes to solar energy inputs (
$\Delta SW_\downarrow^{TOA}$ ) or from internal perturbations within the Earth system ($\Delta\frac{SW_\uparrow^{TOA}}{SW_\downarrow^{TOA}}$). The latter can
be brought about by changes to the reflective properties of Earth's surface which is the focus
of this paper.

 *a. GCM-based radiative kernels*

The radiative kernel technique was developed as a way to assess various climate feedbacks
from climate change simulations across multiple climate models in a computationally efficient
manner (Shell et al., 2008; Soden et al., 2008). A radiative kernel is defined as the differential
response of an outgoing radiation flux at TOA to an incremental change in some climate state
variable -- such as water vapor, air temperature, or surface albedo (Soden et al., 2008). To
generate a radiative kernel for a change in surface albedo with a GCM, the prescribed surface
albedo change is perturbed incrementally by 1%, and the response by the outgoing shortwave
radiation flux at TOA is recorded:
$$\Delta SW_\uparrow^{TOA} = SW_\uparrow^{TOA}(\alpha_s + \Delta\alpha_s) - SW_\uparrow^{TOA}(\alpha_s) = \frac{\partial SW_\uparrow^{TOA}}{\partial\alpha_s}\Delta\alpha_s \equiv K_{\alpha_s}\Delta\alpha_s \qquad (3)$$
where $SW_\uparrow^{TOA}$ is the outgoing shortwave flux at TOA and $K_{\alpha_s}$ is the radiative kernel (in Wm⁻
²) which can then be used with Eq. (1) to estimate an instantaneous shortwave radiative
forcing ($\Delta F$) at TOA:
$$F + \Delta F = LW_\uparrow^{TOA} - (SW_\downarrow^{TOA} - SW_\uparrow^{TOA} + K_{\alpha_s}\Delta\alpha_s)$$
$$\Delta F = -K_{\alpha_s}\Delta\alpha_s \qquad (4)$$

To the best of our knowledge, four albedo change kernels have been developed based on the
following GCMs: the Community Atmosphere Model version 3, or CAM3 (Shell et al.,
2008), the Community Atmosphere Model version 5, or CAM5 (Pendergrass et al., 2018), the
European Center and Hamburg model version 6, or ECHAM6 (Block and Mauritsen, 2014),
and the Geophysical Fluid Dynamics Laboratory model version AM2p12b, or GFDL (Soden
et al., 2008). These four GCM kernels vary in their vertical and horizontal resolutions, their
parameterizations of shortwave radiative transfer, and their prescribed atmospheric state
climatologies. These differences are summarized in Table 1. Apart from differences in their
prescribed atmospheric background states and radiative transfer schemes, a major source of
uncertainty in GCM-based kernels is related to the GCM representation of atmospheric liquid
water/ice associated with convective clouds; of the four aforementioned GCMs, only CAM5
and GFDL attempt to model the effects of convective core ice and liquid in their radiation
calculations (Li et al., 2013).

< Table 1 >

*b. Single-layer atmosphere models of shortwave radiation transfer*
Within the atmospheric science community, various simplified analytical or semi-empirical
modeling frameworks have been developed, either to diagnose effective surface and
atmospheric optical properties from climate model outputs, or to study the relative
contributions of changes to these properties on shortwave flux changes at the top and bottom
of the atmosphere (Atwood et al., 2016; Donohoe and Battisti, 2011; Kashimura et al., 2017;
Qu and Hall, 2006; Rasool and Schneider, 1971; Taylor et al., 2007; Winton, 2005; Winton,
2006). While these frameworks all treat the atmosphere as a single layer, they differ by
whether or not the reflection and transmission properties of this layer are assumed to have a
directional dependency (Stephens et al., 2015) and by whether or not inputs other than those
derived from the boundary fluxes are required (e.g. cloud properties; (Qu and Hall, 2006)).
Winton (2005) presented a semi-empirical four-parameter optical model to account for the
directional dependency of up- and downwelling shortwave fluxes through the one-layer
atmosphere and found good agreement (rRMSE < 2% globally) when benchmarked to online
radiative transfer calculations. Also considering a directional dependency of the atmospheric
optical properties, Taylor et al. (2007) presented a two-parameter analytical model where
atmospheric absorption was assumed to occur at a level above atmospheric reflection. The
analytical model of Donohoe and Battisti (2011) subsequently relaxed the directional
dependency assumption and found the atmospheric attenuation of the surface albedo
contribution to planetary albedo to be 8% higher than the model of Taylor et al. (2007).
Elsewhere, Qu & Hall (2006) developed an analytical framework making use of additional
atmospheric properties such as cloud cover fraction, cloud optical thickness, and the clear-sky
planetary albedo, which proved highly accurate when model estimates of planetary albedo
were evaluated against climate models and satellite-based datasets.
*c. Simple empirical parameterizations of the LULCC science community*
Two simple empirical parameterizations of shortwave radiative transfer have been widely
applied within the LULCC science community for estimating $\Delta F$ from $\Delta \alpha_s$ (Bozzi et al.,
2015; Caiazzo et al., 2014; Carrer et al., 2018; Cherubini et al., 2012; Lutz et al., 2015;
Muñoz et al., 2010). While these parameterizations are also based on a single-layer
atmosphere model of shortwave radiative transfer, at the core of these parameterizations is the
fundamental assumption that radiative transfer is wholly independent of (or unaffected by)
$\Delta \alpha_s$. In other words, they neglect the change in the attenuating effect of multiple reflections
between the surface and the atmosphere that accompanies a change to the surface albedo.
Nevertheless, due to their simplicity and ease of application they continue to be widely
employed in climate research.
**3. Kernel model candidates**
The six candidate models (or parameterizations) for a CERES-based albedo change kernel
(CACK) are presented henceforth. All requisite variables and their derivatives may be
obtained directly from the CERES EBAF v4 products (at monthly and $1° \times 1°$ resolution) and
are presented in Table 2. To improve readability, temporal and spatial indexing is neglected
and all terms presented henceforth in Section 3 denote the monthly pixel means.
< Table 2 >
*a. Analytical kernels*
The first kernel candidate may be analytically-derived from the CERES EBAF all-sky
boundary fluxes and their derivatives. The surface contribution to the outgoing shortwave
flux at TOA $SW_{\uparrow,SFC}^{TOA}$ can be expressed (Donohoe and Battisti, 2011; Stephens et al., 2015;
Winton, 2005) as:
$$SW_{\uparrow,SFC}^{TOA} = SW_{\downarrow}^{TOA} \alpha_s \frac{(1-r-a)^2}{(1-r\alpha_s)} \tag{5}$$

where $r$ is a single pass atmospheric reflection coefficient, $a$ is a single pass atmospheric
absorption coefficient, $SW_{\downarrow}^{TOA}$ is the extraterrestrial (downwelling) shortwave flux at TOA,
and $\alpha_s$ is the surface albedo (defined in Table 2). The expression in the denominator of the
righthand term represents a fraction attenuated by multiple reflections between the surface
and the atmosphere. This model assumes that the atmospheric optical properties $r$ and $a$ are
insensitive to the origin and direction of shortwave fluxes – or in other words – that they are
isotropic.
The single-pass reflectance coefficient is calculated from the system boundary fluxes (Table
2) following Winton (2005) and Kashimura *et al.* (2017):
$$r = \frac{SW_{\downarrow}^{TOA} SW_{\uparrow}^{TOA} - SW_{\downarrow}^{SFC} SW_{\uparrow}^{SFC}}{SW_{\downarrow}^{TOA\ 2} - SW_{\uparrow}^{SFC\ 2}} \tag{6}$$

while the single-pass absorption coefficient $a$ is given as:
$$a = 1 - r - T(1 - \alpha_s r) \tag{7}$$

where $T$ is the clearness index (defined in Table 2). Our interest is in quantifying the $SW_{\uparrow,SFC}^{TOA}$
response to an albedo perturbation at the surface – or the partial derivative of $SW_{\uparrow,SFC}^{TOA}$ with
respect to $\alpha$ in Eq. (5):
$$\frac{\partial SW_{\uparrow}^{TOA}}{\partial \alpha_s} \Delta \alpha_s = K_{\alpha_s}^{ISO} \Delta \alpha_s = \frac{SW_{\downarrow}^{TOA}(1-r-a)^2}{(1-r\alpha_s)^2} \Delta \alpha_s \tag{8}$$
where $K_{\alpha_s}^{ISO}$ is referred to henceforth as the *Isotropic* kernel.
The second analytical kernel is based on the model of Qu and Hall (2006) which makes use of
auxiliary cloud property information commonly provided in satellite-based products of
Earth's radiation budget – including CERES EBAF – such as cloud cover area fraction, cloud
visible optical depth, and clear-sky planetary albedo. This model links all-sky and clear-sky
effective atmospheric transmissivities of the earth system through a linear coefficient $k$
relating the logarithm of cloud visible optical depth to the effective all-sky atmospheric
transmissivity:
$$k = \frac{(T_{a,CLR})-(T_a)}{\ln(\tau+1)} \tag{9}$$
where $T_{a,CLR}$ is the clear-sky effective system transmissivity, $T_a$ is the all-sky effective system
transmissivity, and $\tau$ is the cloud visible optical depth. This linear coefficient can then be
used together with the cloud cover area fraction to derive a shortwave kernel based on the
model of Qu and Hall (2006) – or $K_{\alpha_s}^{QH06}$:
$$\frac{\partial SW_{\uparrow}^{TOA}}{\partial \alpha_s} \Delta \alpha_s = K_{\alpha_s}^{QH06} \Delta \alpha_s = SW_{\downarrow}^{SFC} \left[ (T_a) - kc \ln(\tau+1) \right] \Delta \alpha_s \tag{10}$$
where $c$ is the cloud cover area fraction.
*b. Semi-empirical kernel*
The third kernel makes use of three directionally-dependent (anisotropic) bulk optical
properties $r_\uparrow$, $t_\uparrow$, and $t_\downarrow$, where the first is the atmospheric reflectivity to upwelling
shortwave radiation and the latter two are the atmospheric transmission coefficients for
upwelling and downwelling shortwave radiation, respectively (Winton, 2005).   It is not
possible to derive $r_\uparrow$ analytically from the all-sky boundary fluxes; however, Winton (2005)
provides an empirical formula relating upwelling reflectivity $r_\uparrow$ to the ratio of all-sky to clear-
sky fluxes incident at surface:
$$r_\uparrow = 0.05 + 0.85 \left( 1 - \frac{SW_\downarrow^{SFC}}{SW_{\downarrow,CLR}^{SFC}} \right) \tag{11}$$
where $SW_{\downarrow,CLR}^{SFC}$ is the clear-sky shortwave flux incident at the surface.
Knowing $r_\uparrow$, we can then solve for the two remaining optical parameters needed to obtain our
kernel:
$$t_\downarrow = \frac{SW_\downarrow^{SFC} - r_\uparrow SW_\uparrow^{SFC}}{SW_\downarrow^{TOA}} \tag{12}$$
$$t_\uparrow = T_a - \left[ t_\downarrow - t_\downarrow (1 - r_\uparrow \alpha_s) \right] \tag{13}$$
where $T_a$ is the effective atmospheric transmittance (Table 2) of the earth system.
The kernel may now be expressed as:
$$\frac{\partial SW_\uparrow^{TOA}}{\partial \alpha_s} \Delta \alpha_s = K_{\alpha_s}^{ANISO} \Delta \alpha_s = \frac{SW_\downarrow^{TOA} t_\downarrow t_\uparrow}{(1 - r_\uparrow \alpha_s)^2} \Delta \alpha_s \tag{14}$$
where $K_{\alpha_s}^{ANISO}$ is henceforth referred to as the *Anisotropic* kernel.
*c. Existing empirical parameterizations*
Although not referred to as "kernels" in the literature *per se*, we present the simple empirical
parameterizations as such to ensure consistency with previously described notation and
terminology.

The first candidate parameterization, originally presented in Muñoz *et al.* (2010), makes use
of a local two-way transmittance factor based on the local clearness index:
$$\frac{\partial SW_\uparrow^{TOA}}{\partial \alpha_s}\Delta\alpha_s \equiv K_{\alpha_s}^{M10}\Delta\alpha_s = SW_\downarrow^{TOA}T^2\Delta\alpha_s \qquad (15)$$
where $SW_\downarrow^{TOA}$ is the local incoming solar flux at TOA, $T$ is the local clearness index, and
$\partial SW_\uparrow^{TOA}/\partial\alpha_s$ is the approximated change in the upwelling shortwave flux at TOA due to a
change in the surface albedo.
The second candidate parameterization, originally proposed in Cherubini *et al.* (2012), makes
direct use of the solar flux incident at the surface $SW_\downarrow^{SFC}$ combined with a one-way
transmission constant *k*:
$$\frac{\partial SW_\uparrow^{TOA}}{\partial \alpha_s}\Delta\alpha_s \equiv K_{\alpha_s}^{C12}\Delta\alpha_s = SW_\downarrow^{SFC}k\Delta\alpha_s \qquad (16)$$
where *k* is based on the global annual mean share of surface reflected shortwave radiation
exiting a clear-sky (Lacis and Hansen, 1974; Lenton and Vaughan, 2009) and is hence
temporally and spatially invariant. This value – or 0.85 -- is similar to the global mean ratio
of forward-to-total shortwave scattering reported in Iqbal (1983). Bright & Kvalevåg (2013)
evaluated Eq. (16) at several global locations and found large biases for some regions and
months, despite good overall performance globally (rRMSE $= 7\%$; $n = 120$ months).
*d. Proposed empirical parameterization*
To determine whether the GCM-based kernels could be approximated with sufficient fidelity
using other simpler model formulations based on their own boundary data, we applied
machine learning to identify potential model forms using GCM shortwave boundary fluxes as
input.  For the two GCMs kernels in which the GCM's own shortwave boundary fluxes are
also made available (CAM5 and ECHAM6), we used machine learning to minimize the sum
of squared residuals between the four shortwave boundary fluxes (i.e., $SW_\downarrow^{SFC}$, $SW_\downarrow^{TOA}$,
$SW_\uparrow^{SFC}$, $SW_\uparrow^{TOA}$) and the GCM kernel at the monthly time step.  The reference dataset
consisted of a random global sample of 200,000 monthly kernel grid cells at native model
resolution (97% and 32% of all cells for ECHAM6 and CAM5, respectively) of which 50%
were used for training and 50% for validation.  Models were identified using a form of genetic
programming known as symbolic regression (Eureqa®; Nutonian Inc.; (Schmidt and Lipson,
2009, 2010)) which searches a wide space of model structures as constrained by user input.
In our case, we allowed the model to include the operators (i.e., addition, subtraction,
multiplication, division, sine, cosine, tangent, exponential, natural logarithm, factorial, power,
square root), but numerical coefficients were forbidden.  The model search was allowed to
continue until the percent convergence and maturity metrics exceeded 98% and 50%,
respectively, at which point more than $1 \times 10^{11}$ formulae had been evaluated.  A parsimonious
solution was chosen by minimizing the error metric and model complexity using the Pareto
front (Figure S1 of Supporting Information) (Smits and Kotanchek, 2005).  Between CAM5
and ECHAM6, four common model solutions were found (Table S1 of Supporting
Information).  The best of these common solutions is subsequently referred to as $K_{\alpha_s}^{BO18}$ and is
given as:
$$\frac{\partial SW_\uparrow^{TOA}}{\partial \alpha_s} \Delta\alpha_s = K_{\alpha_s}^{BO18} \Delta\alpha_s = SW_\downarrow^{SFC} \sqrt{T} \Delta\alpha_s \qquad (17)$$


**4. Kernel model evaluation**

*a. Initial candidate screening*

The four GCM kernels presented in Section 2.a are employed as benchmarks to initially screen the six simple model candidates introduced from Section 3b to 3d. We compute a skill metric analogous to the "relative error" metric used to evaluate GCMs by Anav et al. (2013) that takes into account error in the spatial pattern between a model and an observation. Because we have no true observational reference, our evaluation instead focuses on the disagreement or deviation between CERES and GCM kernels at the monthly time step. Given interannual climate variability in the earth system, the challenge of comparing the multi-year CERES kernel to a single-year GCM kernel can be partially overcome by averaging the four GCM kernels.

Using the multi-GCM mean as the reference, we first compute the absolute deviation $AD_{m,p}^{X}$ as:

$$AD_{m,p}^{X} = \left| CERES_{m,p}^{X} - \overline{GCM}_{m,p} \right| \tag{18}$$

where $CERES_{m,p}^{X}$ is the kernel for CERES model candidate $x$ in month $m$ and pixel $p$ and $\overline{GCM}_{m,p}$ is the multi-GCM mean of the same pixel and month. $AD_{m,p}^{X}$ is then normalized to the maximum absolute deviation of all six CERES kernels for the same pixel and month to obtain a normalized absolute deviation, $NAD_{m,p}^{X}$, which is analogous to the "relative error" metric of Anav et al. (2013) having values ranging between 0 and 1:

$$NAD_{m,p}^{X} = 1 - \frac{AD_{m,p}^{X}}{\max(AD_{m,p})} \tag{19}$$

where $\max(AD_{m,p})$ is the maximum absolute deviation of all six CERES kernels at pixel $p$
and month $m$.
CERES kernel ranking is based on the mean relative absolute deviation in both space and time
– or $NAD^X$ :
$$NAD^X = \frac{1}{M}\sum_{m=1}^{M}\frac{1}{P}\sum_{p=1}^{P}NAD_{m,p}^X \qquad (20)$$
where $M$ is the total number of months (i.e., 12) and $P$ is the total number of grid cells.

*b. GCM kernel emulation*
In order to eliminate any bias related to differences in the atmospheric state embedded in the
GCM kernel input climatologies, we emulate them by applying the top candidate models (as
identified from the initial performance screening described in section 4a) using the original
GCM boundary fluxes as input.  Emulation is only done for two of GCM-based kernels since
only two of them have provided the accompanying boundary fluxes needed to do so:
ECHAM6 (Block and Mauritsen, 2014) and CAM5 (Pendergrass et al., 2018).  Emulation
enables a more critical evaluation of the functional form of the candidate models in relation to
the more sophisticated radiative transfer schemes employed by ECHAM6 (Stevens et al.,
2013) and CAM5 (Hurrell et al., 2013).
*c. CACK model uncertainty*
Following emulation, monthly GCM kernels are then regressed on the monthly kernels
emulated with the leading model candidates.  The model that best emulates both GCM kernels
– as measured in terms of the mean coefficient of determination ($R^2$) and mean RMSE – is
chosen to represent CACK.
Three sources of uncertainty are considered for CACK when based on the CERES boundary
flux climatology (i.e., 2001-2016 monthly means): 1) *physical variability* 2) *data uncertainty*;
and 3) *model error* (Mahadevan and Sarkar, 2009).  The first is related to the interannual
variability of Earth's atmospheric state and boundary radiative fluxes.  The second is related
to the uncertainty of the CERES EBAF v4 variables used as input to CACK (including
measurement error).  The third source of uncertainty is the error related to CACK's model
form.  CACK's combined uncertainty for any given pixel and month is estimated as follows,
where if CACK or $y$ is some non-linear function of the CERES boundary inputs $x_1$ and $x_2$
that co-vary in time and space, then the combined uncertainty of $y$ – or $\sigma(y)$ – may be
expressed as the sum of the *model error* plus the combined *physical variability* and *data*
*uncertainty* associated with $x_1$ and $x_2$ summed in quadrature (Breipohl, 1970; Clifford, 1973;
Green et al., 2017):
$$\sigma(y) \approx \sigma_{ME}(y) + \sqrt{\left(\frac{\partial y}{\partial x_1}\right)^2 \left[\sigma_{PV}(x_1) + \sigma_{DU}(x_1)\right]^2 + \left(\frac{\partial y}{\partial x_2}\right)^2 \left[\sigma_{PV}(x_2) + \sigma_{DU}(x_2)\right]^2 + \sqrt{\left(2\frac{\partial y}{\partial x_1}\frac{\partial y}{\partial x_2}\sigma(x_1,x_2)\right)^2}} \quad (21)$$
where $\sigma_{PV}(x_1)$ and $\sigma_{PV}(x_2)$ are the standard deviations of the 16-yr. climatological record of
CERES input variables $x_1$ and $x_2$, respectively, for a given grid cell and month, $\sigma_{DU}(x_1)$ and
$\sigma_{DU}(x_2)$ are the absolute uncertainties of CERES input variables $x_1$ and $x_2$, respectively, for
a given grid cell and month, $\sigma(x_1,x_2)$ is the covariance within the 16-yr. climatological
record between CERES input variables $x_1$ and $x_2$ for a given month and grid cell, and $\sigma_{ME}$ is
the monthly grid cell model error.  Model error ($\sigma_{ME}(y)$) and data uncertainties ($\sigma_{DU}(x_n)$) for
any given grid cell and month are based on the relative RMSE (Supporting Information) and
relative uncertainties of CERES boundary terms reported in Kato *et al.* (2018) (cf. Table 8,
"Monthly gridded, Ocean + Land")  and Loeb *et al.* (2017) (cf. Table 8, "All-sky, *Terra-Aqua*
period"). For the model error, we take the mean relative RMSE of the machine learning
model solutions for ECHAM5 and CAM5. For the relative uncertainty of the incoming solar
flux at TOA ($SW_{\downarrow}^{TOA}$), we use the 1% "calibration uncertainty" reported in Loeb *et al.* (2017).
If CACK's intended application is to estimate a temporally-explicit *ΔF* within the CERES era
(i.e., if temporally-explicit rather than the climatological mean CERES boundary fluxes are
desired to compute CACK), the uncertainty related to *physical variability* ($\sigma_{PV}(x_n)$) can be
dropped from Eq. (21).
*d. Climatological CACK example application*
To demonstrate CACK's application when based on monthly CERES EBAF climatology,
including the handling of uncertainty, we estimate the annual mean local *ΔF* from a $\Delta\alpha$
scenario associated with hypothetical deforestation in the tropics, where *ΔF* for a given month
is estimated as Eq. (4) where $K_{\alpha_s}$ is the 2001-2016 monthly climatological CACK and $\Delta\alpha$ is
the difference in the 2001-2011 monthly climatological mean white-sky surface albedo
between "Croplands" (CRO) and "Evergreen broadleaved forests" (EBF) taken from Gao *et*
*al.* (2014) which is based on International Geosphere-Biosphere Program definitions of land
cover classification.
The monthly climatological albedo look-up maps of Gao *et al.* (2014) contain their own
uncertainties, which we take as the mean absolute difference between the monthly albedos
reconstructed using their look-up model and the monthly MODIS retrieval record (c.f. Table 3
in Gao *et al.* (2014)).
The total estimated uncertainty linked to the annual local (i.e., grid cell) instantaneous *ΔF* can
thus be expressed (in W m$^{-2}$) as:

$$\sigma(\Delta F) = \frac{1}{12} \sum_{m=1}^{12} |\Delta F_m| \sqrt{\left(\frac{\sigma(K_{\alpha_s,m})}{K_{\alpha_s,m}}\right)^2 + \left(\frac{\sigma(\Delta\alpha_{s,m})}{\Delta\alpha_{s,m}}\right)^2} \tag{22}$$

where $\sigma(K_{\alpha_s,m})/K_{\alpha_s,m}$ is the relative grid cell uncertainty of CACK and $\sigma(\Delta\alpha_{s,m})/\Delta\alpha_{s,m}$ is
the relative uncertainty of $\Delta\alpha_s$ in month $m$ defined as:

$$\frac{\sigma(\Delta\alpha_{s,m})}{\Delta\alpha_{s,m}} = \sqrt{\left(\frac{\sigma(\alpha_{s,m})}{\alpha_{CRO,m}}\right)^2 + \left(\frac{\sigma(\alpha_{s,m})}{\alpha_{EBF,m}}\right)^2} \tag{23}$$

where $\sigma(\alpha_{s,m})$ is the monthly absolute uncertainty of the climatological mean surface albedo
(i.e., of the Gao *et al.* (2014) product).

*e. Temporally-explicit CACK application example*

Use of a temporally-explicit CACK may be desirable for time-sensitive applications within
the CERES era. This is particularly true for regions experiencing significant changes to the
atmospheric state affecting shortwave radiation transfer. A good example is in southern
Amazonia where tropical deforestation has been linked to changes in cloud cover (Durieux et
al., 2003; Lawrence and Vandecar, 2014; Wright et al., 2017). To exemplify this, we estimate
the annual mean instantaneous *ΔF* for CERES grid cells in the region having experienced both
significant positive trends in surface albedo and negative trends in cloud area fraction during
the 2001-2016 period. Grid cell trends in surface albedo and cloud area fraction are deemed
significant if the slopes of linear fits obtained from local (i.e., grid cell) ordinary least squares
regressions had p-values $\leq 0.05$. We then apply the slope of the surface albedo trend to
represent the monthly mean interannual $\Delta\alpha$ incurred over the time series together with
CACK updated monthly to estimate the local annual mean instantaneous *ΔF* at each step in
the series:
$$\Delta F(t) = \sum_{m=1}^{m=12} -K_{\alpha_s,m}(t)\Delta\alpha_s \tag{24}$$
where $K_{\alpha_s,m}(t)$ is the monthly CACK in year $t$ of the time series. $\Delta F$ is then averaged across
all grid cells in the sample, with the results then compared to the $\Delta F$ that is computed for the
same grid sample using the time-insensitive CAM5 and ECHAM6 kernels (i.e., $K_{\alpha_s,m} \neq f(t)$).
Using the slope of the surface albedo trend as the $\Delta\alpha_s$ for all months and years rather than the
actual $\Delta\alpha_{s,m}(t)$ (i.e., $\Delta\alpha_{s,m}(t) = \alpha_{s,m,t} - \alpha_{s,m,t-1}$) yields the same result when averaged over the
full time period but allows us to isolate the effect of the changing atmospheric state on
calculations of $\Delta F$. We limit the $\Delta F$ uncertainty estimate to CACK's uncertainty that includes
$\sigma_{DU}(x_n)$ and $\sigma_{ME}(x_n)$ but excludes $\sigma_{PV}(x_n)$.
**5. Results**
*a. Initial performance screening*
Seasonally, differences in latitude band means between the CERES kernel candidates and the
multi-GCM mean kernels are shown in Figure 1.

< Figure 1 >

Qualitatively, starting with December-January-February (*DJF*), $K_{\alpha_s}^{BO18}$ gives the best
agreement with $K_{\alpha_s}^{\overline{GCM}}$ with the exception of the zone around $55 - 65°$S (-55 – -65°), where
$K_{\alpha_s}^{QH06}$ gives slightly better agreement (Fig. 1A). In March-April-May (*MAM*), $K_{\alpha_s}^{BO18}$ appears
to give the best overall agreement with the exception of the high Arctic, where $K_{\alpha_s}^{ANISO}$ and
$K_{\alpha_s}^{C12}$ give better agreement, and with the exception of the zone around $60 - 65°$S (-60 – -65°)
where $K_{\alpha_s}^{QH06}$, $K_{\alpha_s}^{ANISO}$, and $K_{\alpha_s}^{C12}$ agree best with $K_{\alpha_s}^{\overline{GCM}}$ (Fig. 1B). The largest spread in
disagreement across all six CERES kernels is found in June-July-August (*JJA*; Fig. 1 C) at
northern high latitudes. $K_{\alpha_s}^{BO18}$ appears to agree best both here and elsewhere with the
exception of the zone between ~20 – 35°N, where $K_{\alpha_s}^{QH06}$ gives slightly better agreement.
In September-October-November (*SON*), $K_{\alpha_s}^{BO18}$ agrees best with $K_{\alpha_s}^{\overline{GCM}}$ at all latitudes except
the zone between 10 – 25°N and 55 – 65°S where $K_{\alpha_s}^{QH06}$ agrees slightly better.
Quantitatively, the proportion of the total variance explained by linear regressions of monthly
$K_{\alpha_s}^{\overline{GCM}}$ on monthly $K_{\alpha_s}^{CERES}$ (i.e., "$R^2$") is highest and equal for the CERES kernels based on the
ANISO, QH06, and BO18 models (Fig. 2 B, C, & D). Of these three, $K_{\alpha_s}^{QH06}$ has a y-intercept
("$B_0$") closest to 0 and a slope ("$m$") of 1, although the root mean squared error ("*RMSE*") –
an accuracy measure – is slightly better (lower) for $K_{\alpha_s}^{BO18}$. The two CERES kernels with the
lowest $R^2$, highest slopes (negative deviations), highest *RMSE*s, and y-intercepts with the
largest absolute difference from zero – or the worst performing candidates – are those based
on the ISO and M10 models (Fig. 2 A&E).

< Figure 2 >

Although the y-intercept deviation from 0 for $K_{\alpha_s}^{C12}$ is relatively low, its *RMSE* is ~50%
higher than that of $K_{\alpha_s}^{QH06}$, $K_{\alpha_s}^{BO18}$, and $K_{\alpha_s}^{ANISO}$ and leads to notable positive deviation from the
multi-GCM mean ($K_{\alpha_s}^{\overline{GCM}}$) judging by its slope of 0.92.
Globally, *NAD* for the QH06, ANISO, and BO18 kernels are far superior to the ISO, M10,
and C12 kernels (Table 3).

< Table 3 >

After filtering to remove grid cells for oceans and other water bodies, *NAD* scores for these
three kernels decreased; the decrease was smallest for $K_{\alpha_s}^{BO18}$ (-0.03) and largest for $K_{\alpha_s}^{QH06}$ (-
0.06). Despite constraining the analysis to land surfaces only, the rank order remained
unchanged (Table 3), and $K_{\alpha_s}^{QH06}$, $K_{\alpha_s}^{BO18}$, and $K_{\alpha_s}^{ANISO}$ are subjected to further evaluation.
*b. GCM kernel emulation and additional performance evaluation*
However, because the QH06 model ( $K_{\alpha_s}^{QH06}$ ) required auxiliary inputs for cloud cover area
fraction and cloud optical depth – two atmospheric state variables not provided with the
ECHAM6 and CAM5 kernel datasets – it was not possible to emulate these two GCM kernels
with $K_{\alpha_s}^{QH06}$. Additional performance evaluation through GCM kernel emulation is therefore
restricted to the ANISO and BO18 models.
< Figure 3 >
Globally, the kernel based on the ANISO model displays larger annual mean biases relative to
BO18 when compared to both ECHAM6 and CAM5 kernels (Figure 3). Notable positive
biases over land with respect to both ECHAM6 and CAM5 kernels are evident in the northern
Andes region of South America, the Tibetan plateau, and the tropical island region comprising
Indonesia, Malaysia, and Papua New Guinea (Fig. 3 A & C). Notable negative biases over
land with respect to both ECHAM6 and CAM5 kernels are evident over Greenland,
Antarctica, northeastern Africa, and the Arabian Peninsula (Fig. 3 A & C).
< Figure 4 >
Globally, annual biases for BO18 are generally found to be lower than for ANISO and are
mostly non-existent in extra-tropical ocean regions (Fig. 3 B & D). Patterns in biases over
land are mostly negative with the exception of Saharan Africa where the annual mean bias
with respect to both GCMs is positive. For BO18, systematic positive biases – or biases
evident with respect to both GCM kernels – appear over eastern tropical and subtropical
marine coastal upwelling zones where marine stratocumulus cloud dynamics are difficult for
GCMs to resolve (Bretherton et al., 2004; Richter, 2015).
< Table 4 >
Regression statistics (Figure 4) indicate a greater overall performance for BO18 than for
ANISO.  RMSEs for monthly kernels emulated with BO18 are 9.0 and 8.2 W m$^{-2}$ for CAM5
and ECHAM6, respectively – which is ~50-60% of the RMSEs emulated with the ANISO
model.  Relative to ANISO, the BO18 model also gives a higher $R^2$, a slope closer to 1, and a
y-intercept closer to zero (Figure 4).  The BO18 model (or parameterization) is therefore
selected for the CERES albedo change kernel (CACK).
Focusing only on the GCM kernels emulated with $K_{\alpha_s}^{BO18}$ henceforth, global mean negative
biases are evident in all months (Table 4), with the largest biases (in magnitude) appearing in
May (-4.4 W m$^{-2}$) and November (-2.5 W m$^{-2}$) for CAM5 and ECHAM6, respectively.  In
absolute terms, largest biases of 8.6 W m$^{-2}$ and 6.8 W m$^{-2}$ appear in June for CAM5 and
ECHAM6, respectively.  Annually, the mean absolute bias for CAM5 and ECHAM6 is 6.8
and 6.1 W m$^{-2}$, respectively – a magnitude which seems remarkably low if one compares this
to the annual mean disagreement (standard deviation) of 33 W m$^{-2}$ across all four GCM
kernels (not shown; for seasonal mean standard deviations see Fig. 1).
*c. CACK uncertainty*
For a kernel based on 2001-2016 monthly mean CERES EBAF climatology, Figure 5
illustrates the contribution of the absolute error related to $K_{\alpha_s}^{BO18}$'s model form (Fig. 5 A,
annual mean) relative to CACK's total absolute uncertainty (Fig. 5 C, annual mean), which
includes the uncertainty surrounding CERES EBAF v4 input variables $SW_\downarrow^{SFC}$ and $SW_\downarrow^{TOA}$
and their interannual variability (Fig. 5 B, annual mean).
< Figure 5 >
Total propagated $\sigma_{pv}$ and $\sigma_{du}$ far exceeds $\sigma_{me}$, is dominated by $\sigma_{du}(SW_\downarrow^{SFC})$ and
$\sigma_{pv}(SW_\downarrow^{SFC})$, and is largest in the Pacific region to the south of the intertropical convergence
zone (ITCZ). Over land, the annual $\sigma_{pv}$ and $\sigma_{du}$ as well as the annual $\sigma_{total}$ are generally
largest in arid or high altitude regions (Fig. 5 B). However, annual CACK values are also
large in these regions reducing the relative uncertainty (Fig. 5 D). The largest relative
uncertainties over land (on an annual basis) – which can approach 50% – are found over
central Europe, northwestern Asia, southeastern China, Andean Chile, and northwestern N.
America (Fig. 5 D).
*d. Climatological CACK application*
When estimated with a CACK based on monthly CERES EBAF climatology, the annual local
$\Delta F$ from $\Delta\alpha_s$ linked to hypothetical deforestation in the tropics is negative in most regions,
approaching -20 W m$^{-2}$ locally in some regions of the Brazilian Cerrado and south of the
Sahel region in Africa (Fig. 6 B). The combined CACK and $\Delta\alpha_s$ uncertainty for these
regions can approach $\pm$ 5 W m$^{-2}$ annually (Fig. 6 C) in regions like the Brazilian Cerrado and
sub-Sahel Africa. Relative to the $\Delta F$ magnitude, however, the largest uncertainties (annual)
may be found in the subtropical regions of Central America, southern Brazil, southern Asia,
and northern Australia, where it can approach 30-40% (Fig. 6 D).
*e. Temporally-explicit CACK application*
The effect of a decreasing cloud cover and increasing surface albedo trend in southern
Amazonia (Fig. 7 B) on shortwave radiative transfer and thus a CACK-based estimate of
regional mean annual $\Delta F$ emerges in Figure 7 C, where $\Delta F$ increases in magnitude by 0.004
W m$^{-2}$ from 2002 to 2016.  This $\Delta F$ trend would otherwise go undetected if a GCM-based
kernel were applied to the same surface albedo trend – that is, to a sustained positive
interannual monthly albedo change "pulse".  Alternatively, a CACK based on 2001 CERES
EBAF inputs (applied with $\Delta\alpha_s$ for 2001-2002) would give slightly higher $\Delta F$ estimates
relative to those based on ECHAM6 and CAM5 kernels; conversely, a CACK based on 2015
CERES EBAF inputs (applied with $\Delta\alpha_s$ for 2015-2016) that would yield lower $\Delta F$ estimates
relative to those based on the same two GCM-based kernels (Fig. 7 C). Use of temporally-
explicit CACK can therefore capture $\Delta F$ trends related to a changing atmospheric state that
fixed-state GCM kernels are unable to capture.
**5. Discussion**
Motivated by an increasing abundance of climate impact research focusing on land processes
in recent years, we comprehensively evaluated six simplified models (or parameterizations) as
candidates for an albedo change kernel based on the CERES EBAF v4 products (Kato et al.,
2018; Loeb et al., 2017). Relative to albedo change kernels based on sophisticated radiative
transfer schemes embedded in GCMs, a CERES-based albedo change kernel – or CACK –
represents a more transparent and empirically-rooted alternative that can be updated
frequently at relatively low cost.  This allows greater flexibility to meet the needs of research
focusing on surface albedo trends within the CERES era in regions currently undergoing rapid
changes to atmospheric state as it affects shortwave radiation transfer.  Although some
modeling groups have provided recent updates to their albedo change kernels using the latest
GCM versions (e.g., (Pendergrass et al., 2018)), the atmospheric state conditions used to
derive them may still be considered outdated or not in sync with that required for many
applications (Table 1).
Based on both qualitative and quantitative benchmarking against the mean of four GCM
kernels, the novel kernel parameterization obtained from machine learning $K_{\alpha_s}^{BO18}$, together
with the two (semi-)analytically derived kernels $K_{\alpha_s}^{QH06}$ and $K_{\alpha_s}^{ANISO}$, proved far superior to the
$K_{\alpha_s}^{ISO}$ analytical kernel and to the two additional empirical parameterizations $K_{\alpha_s}^{C12}$ and $K_{\alpha_s}^{M10}$.
When subjected to additional performance evaluation, however, we found that $K_{\alpha_s}^{BO18}$ was
able to more robustly emulate two GCM kernels (ECHAM6 and CAM5) with exceptionally
high agreement, suggesting that $K_{\alpha_s}^{BO18}$ could serve as a suitable candidate for CACK.
Relative to the monthly CAM5 and ECHAM6 kernels, the mean absolute monthly emulation
"error" of $K_{\alpha_s}^{BO18}$ was found to be 6.8 and 6.1 W m$^{-2}$, respectively – a magnitude which is only
~20% of the standard deviation found across four GCM kernels (annual mean).  CACK's
remarkable simplicity lends support to the idea of using machine learning to explore and
detect emergent properties of radiative transfer or other complex, interactive model outputs in
future research.  The fact that the $K_{\alpha_s}^{BO18}$ parameterization emerged as the best common
solution from two independently executed machine learning analyses each employing a
random sampling unique to a specific GCM kernel suggests that the $K_{\alpha_s}^{BO18}$ parameterization is
robust and insensitive to the underlying GCM representation of shortwave radiative transfer.
Despite its stronger empirical foundation over a GCM-based kernel, it is important to
recognize CACK's limitations.  Firstly, while CACK has a finer spatial resolution than most
GCM kernels, it still represents a spatially averaged response rather than a truly local
response; in other words, the state variables used to define the $SW_{\uparrow}^{TOA}$ response are averages
tied to the coarse spatial (i.e., 1° x 1°) resolution of the CERES EBAF v4 product grids.
Secondly, the monthly CERES EBAF-Surface product used to define lower atmospheric
boundary conditions is not strictly an observation. The space-borne platform is not able to
directly observe surface irradiances, requiring additional satellite-based estimates of cloud and
aerosol properties as input to a radiative transfer model (Kato et al., 2012). Although TOA
irradiances are applied to constrain the surface irradiances, they remain susceptible to errors
in the radiative transfer model inputs. Considering this error as "data uncertainty" increases
CACK's overall uncertainty beyond that which is related to its underlying parameterization or
"model error". The uncertainty of CERES surface shortwave irradiances as well as extensive
ground validation and testing are documented in greater detail elsewhere (Kato et al., 2013;
Kato et al., 2018; Loeb et al., 2017; Loeb et al., 2009) and may continue to be reduced in
future EBAF-Surface versions.
*Concluding remarks*
To conclude, we developed, evaluated, and proposed a radiative kernel for surface albedo
change based on CERES EBAF v4 products – or CACK. Relative to existing kernels based on
GCMs, CACK provides a higher spatial resolution, higher transparency alternative that is
more amenable to user needs. For LULCC research of the near-past, present day, or near-
future periods, application of a CACK whose inputs are based on monthly climatological
means of the full CERES EBAF record can better-account for the corresponding interannual
variability in Earth's atmospheric state affecting shortwave radiative transfer. For regions
undergoing changes in atmospheric state that are detectable above the normal variability
within the CERES era, application of a temporally-explicit CACK can better-account for its
influence on $\Delta F$ estimates from surface albedo change. CACK's input flexibility and
transparency combined with documented uncertainty make it well-suited to be applied as part

of a Monitoring, Reporting, and Verification (MRV) framework for biogeophysical impacts on land, analogous to those which currently exist for land sector greenhouse gas emissions.

**Code and Dataset Availability**

We make both monthly temporally-explicit and monthly climatological mean CACKs for years 2001-2016 available as a complete data product ("CACKv1.0"; (Bright and O'Halloran, 2019)) that includes their respective uncertainty layers. A summary of this dataset and associated variables is provided in Table S3 of the Supporting Information. Octave script files for generating monthly CACK and demonstrating its application with user-specified temporal and spatial extents are bundled with the netCDF file.

**Data Availability**

CERES EBAF data are available for download at: https://ceres.larc.nasa.gov/products.php?product=EBAF-TOA . The CAM3 kernel is available at: http://people.oregonstate.edu/~shellk/kernel.html . The CAM5 kernel is available at: https://www.earthsystemgrid.org/ac/guest/secure/sso.html . The ECHAM5 kernel is available at: https://swiftbrowser.dkrz.de/public/dkrz_0c07783a-0bdc-4d5e-9f3b-c1b86fac060d/Radiative_kernels/ .

**Acknowledgements**

R.M.B. was supported by the Research Council of Norway, grants #244074/E20 and #250113/F20; T.L.O. was supported by the Climate and Land Use program award #2017-68002-26612 of the USDA National Institute of Food and Agriculture.

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

**Table 1.** Attributes of existing GCM kernels, all of which having a monthly temporal
resolution.

| Kernel | Base climatology extent | Base climatology period | Shortwave Radiative transfer | Horizontal Resolution | References |
|---|---|---|---|---|---|
| ECHAM6 | 1,000 years | Preindustrial* | RRTM-G | $1.88° \times 1.88°$ | (Block and Mauritsen, 2014; Stevens et al., 2013) |
| CAM3 | 6 years | 1995-2000 | δ-Eddington | $1.4° \times 1.4°$ | (Collins et al., 2006; Shell et al., 2008) |
| CAM5 | 1 year | 2006-2007 | RRTM-G | $0.94° \times 1.25°$ | (Pendergrass et al., 2018) |
| GFDL | 17 years | 1979-1995 | Exponential sum-fits, 18 bands | $2° \times 2.5°$ | (Soden et al., 2008; The GFDL Global Atmospheric Model Development Team, 2004) |

*Atmospheric $CO_2$ concentration = 284.7 ppmv; Exact time period unknown


**Table 2**. Definition of CERES input variables and other system optical properties derived
from CERES inputs. All variables have a monthly temporal resolution and a spatial
resolution of $1° \times 1°$.

### CERES EBAF v.4 Shortwave Boundary Fluxes

| | | |
|---|---|---|
| $SW_\downarrow^{TOA}$ | Downwelling solar flux at top-of-atmosphere | Wm$^{-2}$ |
| $SW_\downarrow^{SFC}$ | Downwelling solar flux at surface | Wm$^{-2}$ |
| $SW_{\downarrow,CLR}^{SFC}$ | Clear-sky downwelling solar flux at surface | Wm$^{-2}$ |
| $SW_\uparrow^{TOA}$ | Upwelling solar flux at top-of-atmosphere | Wm$^{-2}$ |
| $SW_\uparrow^{SFC}$ | Upwelling solar flux at surface | Wm$^{-2}$ |

### System Optical Properties

| | | |
|---|---|---|
| $T = SW_\downarrow^{SFC} \big/ SW_\downarrow^{TOA}$ | Clearness index | unitless |
| $\alpha_p = SW_\uparrow^{TOA} \big/ SW_\downarrow^{TOA}$ | Planetary albedo | unitless |
| $\alpha_s = SW_\uparrow^{SFC} \big/ SW_\downarrow^{SFC}$ | Surface albedo | unitless |
| $A_p = 1 - \alpha_p$ | Effective planetary absorption | unitless |
| $A_s = \left[ SW_\downarrow^{SFC} - SW_\uparrow^{SFC} \right] \big/ SW_\downarrow^{TOA}$ | Effective surface absorption | unitless |
| $A_a = A_p - A_s$ | Effective atmospheric absorption | unitless |
| $T_a = 1 - A_a$ | Effective atmospheric transmission | unitless |
| $T_{a,CLR} = 1 - A_{a,CLR}$ | Clear-sky effective atmospheric transmission | unitless |
| $\tau$ | Cloud visible optical depth | unitless |
| $c$ | Cloud area fraction | fraction |



**Table 3.** Normalized absolute deviation and CERES kernel model candidate ranking.

|        | Global | | Land only | | |
|--------|--------|------|-----------|------|-----------|
|        | *NAD*  | Rank | *NAD*     | Rank | Mean Rank |
| **ISO**   | 0.05 | 6 | 0.05 | 6 | *6* |
| **ANISO** | 0.64 | 3 | 0.59 | 3 | *3* |
| **C12**   | 0.45 | 4 | 0.47 | 4 | *4* |
| **M10**   | 0.26 | 5 | 0.34 | 5 | *5* |
| **QH06**  | 0.66 | 2 | 0.60 | 2 | *2* |
| **BO18**  | 0.67 | 1 | 0.64 | 1 | *1* |



**Table 4.** Global monthly mean bias (*MB*) and mean absolute bias (*MAB*) for $K_\alpha^{BO18}$ emulated with $T$ and $SW_\downarrow^{SFC}$ from ECHAM6 and CAM5. For reference, the global mean value of $K_\alpha^{BO18}$ is 133 W m$^{-2}$.

|  | \multicolumn{13}{c}{*MB* (W m$^{-2}$)} |
|---|---|---|---|---|---|---|---|---|---|---|---|---|---|
|  | **Jan.** | **Feb.** | **Mar.** | **Apr.** | **May** | **Jun.** | **Jul.** | **Aug.** | **Sep.** | **Oct.** | **Nov.** | **Dec.** | **Ann.** |
| $K_\alpha^{BO18} - K_\alpha^{CAM5}$ | -2.9 | -3.4 | -3.3 | -3.9 | -4.4 | -3.8 | -3.8 | -3.7 | -3.4 | -3.8 | -3.7 | -3.3 | -3.6 |
| $K_\alpha^{BO18} - K_\alpha^{ECHAM6}$ | -1.9 | -2.2 | -1.8 | -1.9 | -2.2 | -1.5 | -1.1 | -1.6 | -1.7 | -2.5 | -2.5 | -1.8 | -1.9 |
|  | \multicolumn{13}{c}{*MAB* (W m$^{-2}$)} |
|  | **Jan.** | **Feb.** | **Mar.** | **Apr.** | **May** | **Jun.** | **Jul.** | **Aug.** | **Sep.** | **Oct.** | **Nov.** | **Dec.** | **Ann.** |
| $\lvert K_\alpha^{BO18} - K_\alpha^{CAM5} \rvert$ | 6.9 | 5.7 | 5.2 | 6.8 | 7.7 | 8.6 | 7.9 | 6.7 | 5.6 | 6.1 | 6.9 | 6.9 | 6.8 |
| $\lvert K_\alpha^{BO18} - K_\alpha^{ECHAM6} \rvert$ | 6.3 | 5.7 | 5.0 | 5.9 | 6.7 | 6.8 | 6.4 | 5.8 | 5.3 | 5.6 | 6.4 | 6.7 | 6.1 |


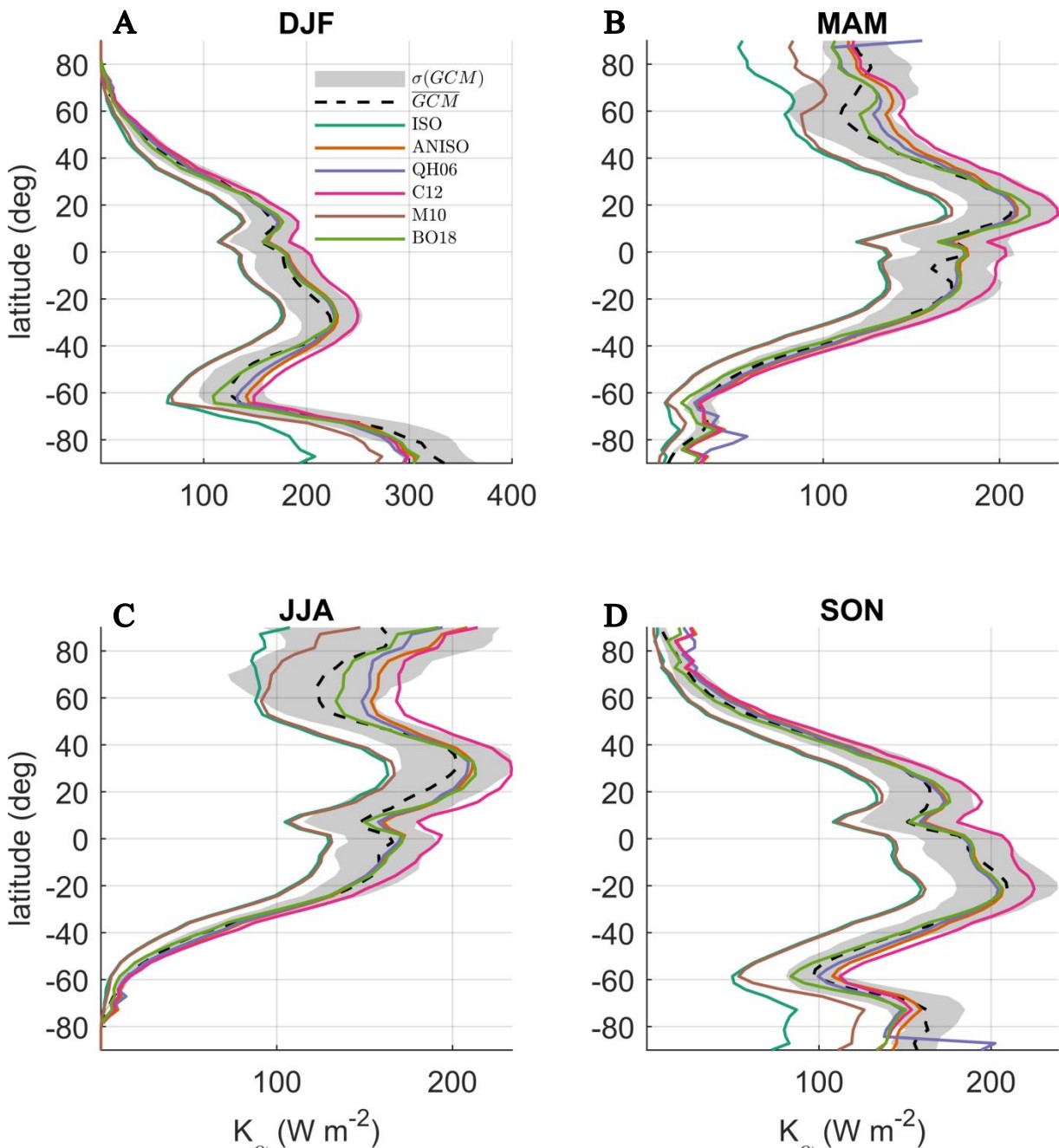

**Figure 1.** Latitudinal (1°) and seasonal means of the multi-GCM mean ($K_\alpha^{\overline{GCM}}$) and CACK model candidates for: A) December-January-February (DJF); B) March-April-May (MAM); C) June-July-August (JJA); D) September-October-November (SON). CACK model candidates refer to those presented in section 3 and not to those of the model selection phase of the machine learning algorithm.

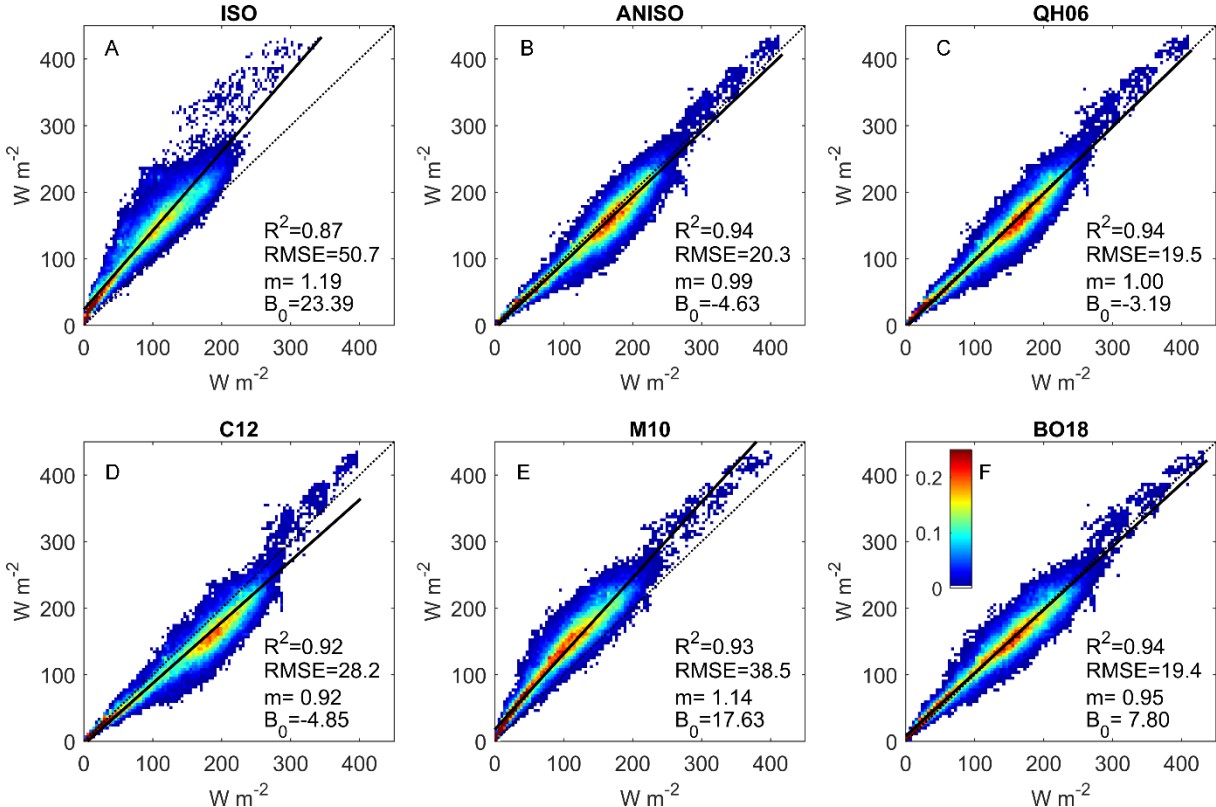

**Figure 2.** A)-F): Scatter-density regressions of global monthly mean $K_\alpha^{\overline{GCM}}$ (y-axis) and $K_\alpha^{CERES}$ (x-axis), with the CERES kernel identifier shown at the top of each sub-panel. "$m$" = slope; "$B_0$" = y-intercept. The color scale indicates the percentage of regression points that fall within an averaging bin, where the x-axis and y-axis have been gridded into $100 \times 100$ equally-spaced bins to help illustrate the density of overlapping points.

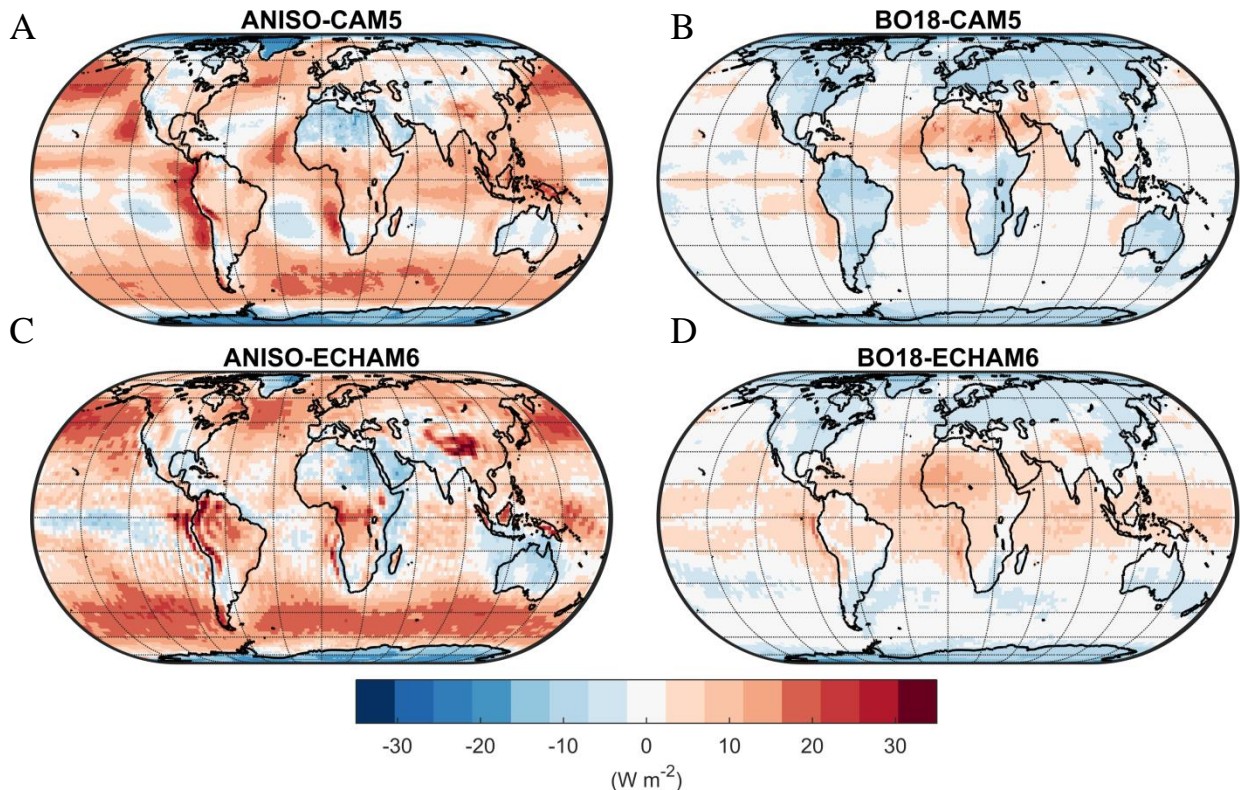

**Figure 3.** A) Mean annual bias of the CAM5 albedo change kernel emulated with the ANISO semi-empirical model; B) Mean annual bias of the CAM5 albedo change kernel emulated with the BO18 parameterization; C) Mean annual bias of the ECHAM6 albedo change kernel emulated with the ANISO semi-empirical model; D) Mean annual bias of the ECHAM6 albedo change kernel emulated with the BO18 parameterization

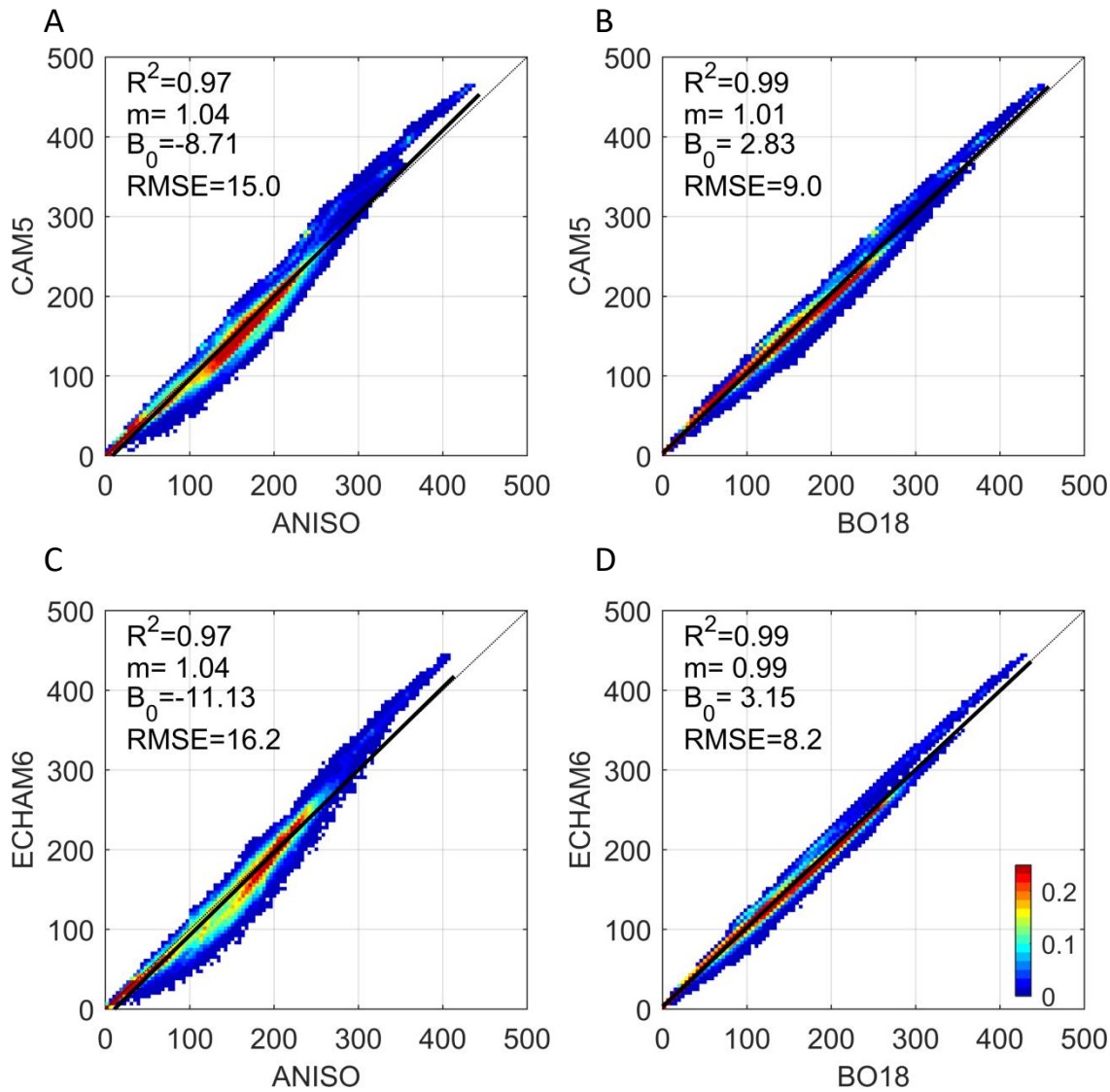

**Figure 4.** A)-D): Scatter-density regressions of $K_\alpha^{GCM}$ (y-axis) and $K_\alpha^{GCM}$ emulated with the ANISO semi-empirical model and BO18 parameterization (x-axis); "$m$" = slope; "$B_0$" = y-intercept. See Figure 2 caption for a description of the color scale.

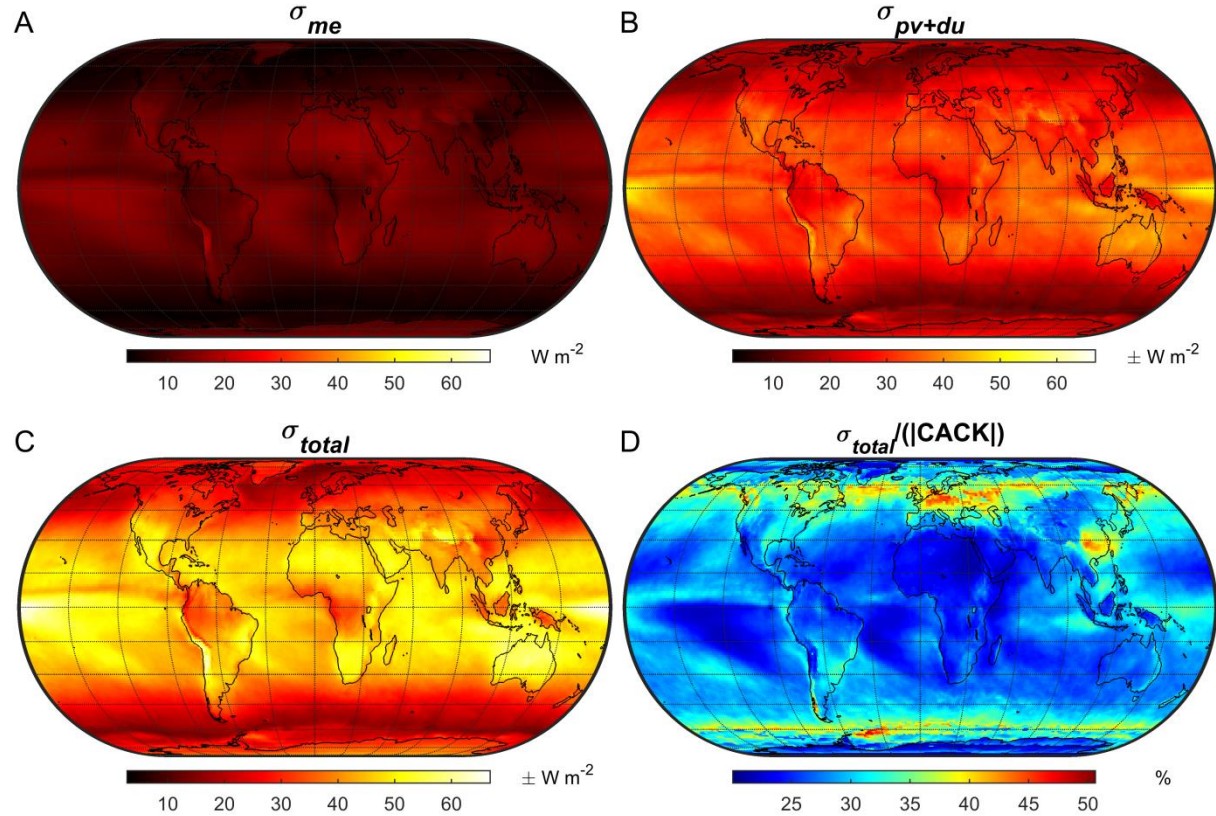

884

**Figure 5.** Annual uncertainty of a CACK based on 2001-2016 monthly mean CERES EBAF

v4 climatology: A) The absolute uncertainty related to *model error* (i.e., the $K_{\alpha_s}^{BO18}$

parameterization); B) The total propagated absolute uncertainty related to *physical variability*

and *data uncertainty* of CACK input variables; C) Total absolute uncertainty; D) Total

relative uncertainty.

890

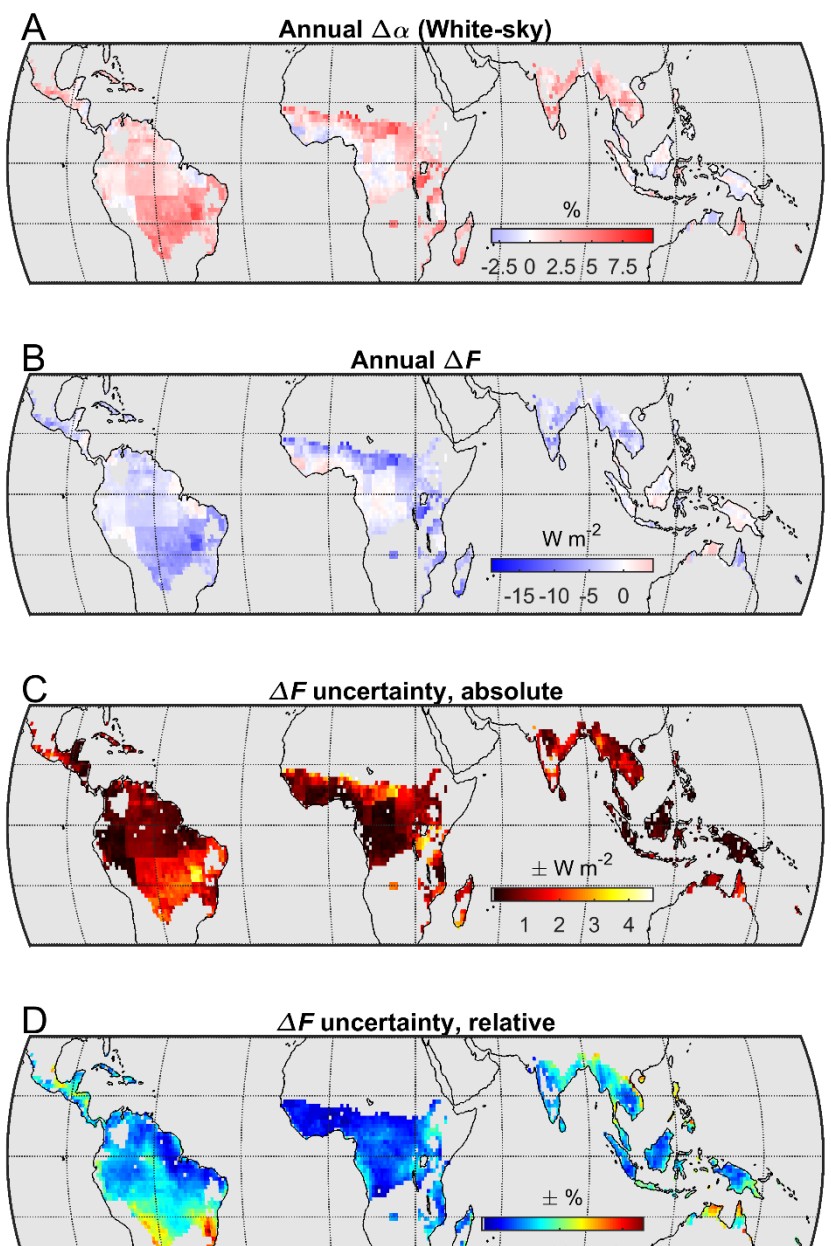

891

**Figure 6.** Example application of a CACK based on the 2001-2016 monthly mean CERES

EBAF v4 climatology to estimate the local annual mean $\Delta F$ from a hypothetical land cover

change within a CERES grid cell.  A)  Annual mean of the climatological (i.e., 2001-2011)

monthly mean difference in white-sky surface albedo between *croplands* and *evergreen*

*broadleaved forests* ($\Delta \alpha_s$) based on the 1° product of Gao *et al*. (2014); B) Annual mean

local (i.e., within grid cell) instantaneous radiative forcing ($\Delta F$) of monthly mean $\Delta \alpha_s$

estimated with CACK; C) Absolute uncertainty (annual mean) of the CACK-based $\Delta F$

899   estimate, including the uncertainty of $\Delta\alpha_s$ ; D) Relative uncertainty (annual mean) of the

900   CACK-based $\Delta F$ estimate.

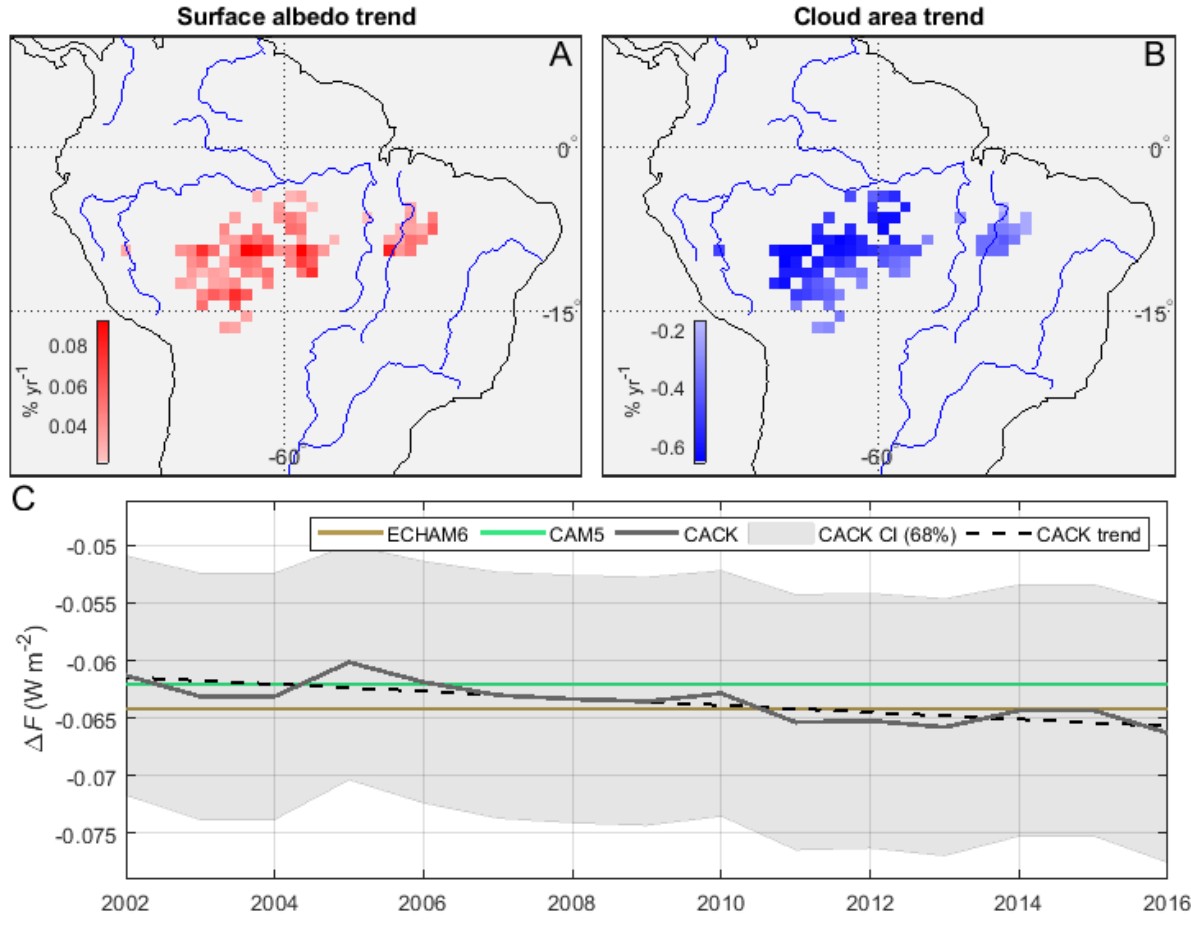

**Figure 7.** Example application of a temporally-explicit CACK. A) 2001-2016 statistically

significant positive trends in all-sky *surface albedo* derived from CERES EBAF-Surface v4;

B) 2001-2016 statistically significant negative trends in *cloud area* derived from CERES

EBAF-TOA v4; C) Mean $\Delta F$ from $\Delta \alpha_s$ when estimated with the CACK, ECHAM6, and

CAM5 surface albedo change kernels. $\Delta F$ is the mean of all grid cells plotted in panel A).

The 1σ confidence interval ("CI") shown for CACK excludes the uncertainty component

related to *physical variability*.