# Peer review of "Developing a monthly radiative kernel for surface albedo change from satellite"

_Geoscientific Model Development, 2019_

## Referee Comment (RC1) · Anonymous Referee #1 · 29 Mar 2019

This study by Bright and O'Halloran developed shortwave radiative kernels based on the CERES EBAF products, which would be an alternative to GCM-based kernels. The performance of the observation-based kernels is also evaluated based on the multi-GCM mean. This is an interesting study, and the developed shortwave radiative kernels have the potential of being used for land use-climate studies. However, I think the manuscript needs some improvement and further development in the analysis before it can be published.

My major concerns include:

[Figure]

1. The evaluation of CERES kernels uses four GCM kernels as benchmarks. I am wondering the uncertainties among the different GCMs. First, why are these four models chosen? But why CAM3 and GFDL are not mentioned in the results? Second, for Figure 1, if plotting the radiative kernel for individual GCMs, is there a large spread like the CERES-based estimates? Third, are the author's conclusions model-dependent? Because the BO18 kernel is trained using the multi-GCM mean as the reference, it is not surprising that it has better performance than other CERES kernels. However, if using a single GCM (or including other GCMs, like HadGEM2 radiative kernels, Smith 2018) as the benchmarks, will QH06 or ANISO still be better than other kernel models? The authors may need more analysis and discussion about the model dependency.

2. One of the motivations of this study is "atmospheric state variables used as model input are limited to single years, thus being sensitive to anomalous weather conditions that may have occurred in those years". Can you explain more about this? As the authors mentioned in L278, they are comparing the multi-year CERES kernel to a single-year GCM kernel. I assume the GCM simulations are only one-year long? The authors may need to provide more description and discussion about these GCM simulations. If the simulates are for a specific year (which year?), or a climatological run, are they comparable to the CERES-based kernel models which are for the period 2001-2016. Additionally, I am curious about the inter-annual variability of the multi-year CERES kernels.

3. This study is started with the "need within LULCC science community for simple and transparent tools for predicting radiative forcings from surface albedo changes". Is it possible to provide a simple example of how to apply CACK v1.0 to the LULCC studies?

Specific comments:

1. The organization of section 2 and section 3 is a little confusing. The title of section 2 is "Review of existing approaches", but most of the kernels described in section 3 are

also "existing approaches", aren't they? 2. L40, What do you mean by "offline"? Run land surface model offline? I also can't find the paper (Randerson et al. 2006) in the reference. 3. L151, Eq. (3) and Eq. (4), are $\Delta\alpha$s and $\Delta\alpha$ the same thing? If yes, it would be better to keep the consistency. 4. L247, Which part (or period) of data is used for model training, and which part is used for prediction? 5. L263, It should be "e. Initial screening of candidate models for a CERES-based kernel" 6. L409, They are mean absolute bias, not RMSD. 7. L441-444, Can the authors explain more about how the land-based solar radiation management is an example of the CACK's flexibility?

Reference:

Smith, Christopher J. (2018) HadGEM2 radiative kernels. University of Leeds. [Dataset] https://doi.org/10.5518/406

---

## Referee Comment (RC2) · Anonymous Referee #2 · 10 Apr 2019

- General comments

The manuscript presented by Bright and O'Halloran suggests the use of a new kernel (CACKv1.0) to derive radiative forcing at the top of the atmosphere from surface albedo changes. This kernel is derived by applying a machine learning technique to identify a formula which can best reproduce the results from kernels derived from Global Circulation Models, once it is applied to CERES satellite-derived data. The authors argue that compared to GCM-derived kernels, this new formula would 1) enable a more transparent derivation of radiative forcing from surface albedo changes, and 2) rely on data from

several years. Their analysis shows that the new formula performs better at mimicking the results from GCM-derived kernels compared to previously suggested formulations. They suggest the use of their results by the scientists studying the impacts of land-use and land-cover changes (LULCC) on climate to improve their calculations of radiative forcing from surface albedo changes.

Having an easily applicable kernel that reproduces the results from GCMs can indeed be useful for the LULCC community, and it that sense the authors' initiative is welcome and scientifically significant. Having said that, there are a couple of issues with the authors' approach, while the methodology could be better described to ensure reproducibility of the results. Overall, substantial work also needs to be done on the writing to improve understandability of the manuscript. These issues are not insurmountable, but I recommend that they are addressed before the manuscript is accepted.

- Specific comments

The real added value of CACK compared to previously suggested simple formulations can only be assessed in light of the uncertainties between GCM kernels. These thus need to be included at least in Figure 1 and discussed in the manuscript, so that the readers can assess for themselves how much of a difference using CACK rather than a simple isotropic kernel (for example) makes. The authors also mention that the GCM-derived kernels are based on single years of forcing data. This renders them uncertain and thus less appropriate as a benchmark, therefore the authors choose to use the multi-GCM mean kernel as a reference to partly alleviate the lack of consideration of interannual variability when they were derived. This seems reasonable but only partly alleviates the issue. In addition to being explicitly shown and discussed, the uncertainties about GCM-derived kernels (both related to model spread and interannual variability) need to be acknowledged in the Discussion.

Even in the current state, more conclusions could be drawn from Figure 1 by describing for example which kernels perform worst against the GCM-derived ones and potentially

advancing reasons why this is the case.

The methodology should be more detailed to be able to understand how Equation 16 is derived. Which optimal structures and coefficients are considered during the symbolic regression? What should make the reader think that this approach doesn't miss potentially relevant formulas? And which "boundary fluxes (or system parameters derived from these fluxes) that minimized the sum of squared residuals…" were considered? This information should at least be provided in the Supplementary Material.

Given that the GCMs and the CERES data are not available at the same resolution, some kind of regridding must have been conducted. Some regridding methods make more sense than others, therefore it would be useful to have some more information on this aspect.

It is also not so clear from the current manuscript why certain choices were made regarding the GCM and kernel selections. Why are four GCM kernels included in the study, are these the only ones available? Is there some information existing on the quality of these kernels that guided the selection? Could the authors justify why they "emulated" the kernels of just two GCMs in a second step? It seems like only the 3 kernels performing best against the GCM-derived ones were retained for further analysis, but this is also not explicitly mentioned.

The structure of the manuscript could be improved to facilitate understandability. For example, why not mentioning the isotropic and anisotropic kernels, as well as the kernel from Qu and Hall in Section 2 already. Currently, at first it may read like they have been derived by the authors. The names of the studies that introduced other types of statistical kernels could also be added in the subsection titles to help the reader follow. The description of the CERES dataset also seems misplaced in Section 2. Additionally, in some occurrences the subsection numbering is wrong and the placeholders for Figures or Tables misplaced.

Last but not least, the CACK dataset is only mentioned in the conclusion, although from

the title it sounds like an important output of the study. If this is the case, it would need to be introduced in the abstract and the introduction of the manuscript. But ultimately, one may wonder whether describing CACK as a dataset is appropriate. Could the authors maybe develop on what makes it more than just applying Eq. 16 to CERES data, for example in terms of pre-processing or perspectives for updates, etc.?

- Technical comments

l. 68: "An additional downside is the that". Check typo

l. 157: to facilitate understandability it could be good to repeat the downsides of GCM-derived kernels here

l. 211 and 230: isn't the subscript "SFC" missing in the left-hand terms?

l. 425: "course" should read "coarse"

l. 704-705: can the authors make clearer what is meant by "100X100 sample grid"?

---

## Author Comment (AC1) · 26 Jun 2019

Dear editors and reviewers,

Our response to two reviewers is the attached ZIP archive, which contains the following files:

1. A cover letter for our response to reviews (.docx file) 2. Two reviewer comments with our responses in red (.docx file) 3. The revised manuscript (without tracked changes, .docx file) 4. A new supplementary Information file (.docx file). 5. The

revised manuscript with changes tracked (.docx file) 6. A PDF of all the previous .docx files concatenated.

Thank you very much.

Tom O'Halloran and Ryan Bright June 25, 2019

Please also note the supplement to this comment: https://www.geosci-model-dev-discuss.net/gmd-2019-15/gmd-2019-15-AC1-supplement.zip

---

## Author Response (AR1)

June 25, 2019

**GMD-2019-15: Revision**

Dear Editor,

We are pleased to submit our revised manuscript entitled: "Developing a monthly radiative kernel for surface albedo change from satellite climatologies of Earth's shortwave radiation budget: CACK v1.0" for publication consideration in *Geoscientific Model Development*.

Major changes to the manuscript include:

- A major re-structuring to improve overall flow and readability. This re-structuring was necessary to showcase CACK v1.0 as a comprehensive, transparent, and flexible dataset built on a novel model (parameterization) of shortwave radiation transfer.
- An expanded analysis of CACK's performance including new content on uncertainty and two new demonstrations of its application
- An improved description of the methods to ensure reproducibility, in particular that pertaining to the symbolic regression analysis
- The addition of a Supporting Information document providing additional detail surrounding CACK's uncertainty calculations, the symbolic regression method and results, and a detailed description of the CACK v1.0 dataset which now includes estimates for three sources of uncertainty.

The revised manuscript has increased by ~2,000 words, 3 figures, and 1 table. We feel confident that our revisions go above and beyond that which is required to satisfy reviewers and add notable value to the paper serving to elevate its overall impact. For instance, the new and comprehensive analysis on uncertainty and its inclusion in CACK v1.0 should make it more attractive as a credible candidate for use as part of a future Monitoring, Reporting, and Verification (MRV) framework for radiative forcing impacts of albedo changes on land.

Please do not hesitate contacting us should you require additional information or clarification.

Kind Regards,

Ryan M. Bright and Tom L. O'Halloran

**Reponses to Anonymous Referee #1**

This study by Bright and O'Halloran developed shortwave radiative kernels based on the CERES EBAF products, which would be an alternative to GCM-based kernels. The performance of the observation-based kernels is also evaluated based on the multi-GCM mean. This is an interesting study, and the developed shortwave radiative kernels have the potential of being used for land use-climate studies. However, I think the manuscript needs some improvement and further development in the analysis before it can be published.

We thank Anonymous Referee #1 for his/her constructive feedback. To address his/her major concerns, we have provided more detail about the GCM kernels and their uncertainties, improved the description of our methodology, and provided two examples illustrating CACK's application.

My major concerns include:

1. The evaluation of CERES kernels uses four GCM kernels as benchmarks. I am wondering the uncertainties among the different GCMs. GCM uncertainties are largely related to their representation of low-level cloud cover and properties (please see our reference to Dolinar *et al.* 2015 [original manuscript P3 L67]). Regarding cloud properties, one of the major differences among GCMs is related to the representation of atmospheric liquid water/ice associated with convective clouds. Of the four GCMs we considered, only CAM5 and GFDL attempt to model the effects of precipitating and/or convective core ice and liquid in their radiation calculations. We add this detail in (new) Section 2.a and provide a new citation (e.g., to Li et al. (2013)). First, why are these four models chosen? These GCM kernels were chosen simply because at the time the study commenced these were the only ones available. We add this rationale to the main text (new Section 2.a). But why CAM3 and GFDL are not mentioned in the results? We carried out a two-stage evaluation, where CAM3 and GFDL comprised part of the "multi-GCM mean" benchmark we used in the first stage (described in new Section 4a), whose results are presented in (new) Section 5, Figures 1 & 2. We hope our re-organization and improved methods descriptions have now made this clearer. Second, for Figure 1, if plotting the radiative kernel for individual GCMs, is there a large spread like the CERES-based estimates? This is a great question and we agree that the spread in GCMs should be made more visible. We have revised Figure 1 such that is now shows the spread (taken as 1 standard deviation) in latitudinal means across the four GCMs. Third, are the author's conclusions model-dependent? Because the BO18 kernel is trained using the multi-GCM mean as the reference, it is not surprising that it has better performance than other CERES kernels. This is a fair comment and valid concern. To check this, we re-ran the machine learning algorithm twice, first using a random sample of the CAM5 kernel (as the dependent) with its own boundary fluxes (as independents), the second time using a random sample of the ECHAM6 kernel with its own boundary fluxes as input (note: these were the only two kernels for which the boundary fluxes used to derive them were also available to us). The BO18 model emerged as the best solution (i.e., model form) common to the two independent machine learning analyses. Because the BO18 model was then applied using CERES EBAF inputs and subsequently compared to a multi-GCM mean that included the two additional GCM kernels (i.e., GFDL and CAM3) that were not part of the model training exercise, we feel confident that the BO18 model is robust and insensitive to the GCM kernels used for training. However, if using a single GCM (or including other GCMs, like HadGEM2 radiative kernels, Smith 2018) as the benchmarks, will QH06 or ANISO still be better than other kernel models? Yes, we indeed found this to be the case – that whether benchmarking to multi-GCM means or to specific GCMs, the CERES kernel performance ranking remained unchanged (excluding the QH06 kernel for the reason provided in revised Section 5b). The authors may need more analysis and discussion about the model dependency. We have added a section in the Discussion regarding BO18's model (in)dependency.

2. One of the motivations of this study is "atmospheric state variables used as model input are limited to single years, thus being sensitive to anomalous weather conditions that may have occurred in those years". Can you explain more about this? As the authors mentioned in L278, they are comparing the multi-year CERES kernel to a single-year GCM kernel. I assume the GCM simulations are only one-year long? The authors may need to provide more description and discussion about these GCM simulations. The GCM simulations from which the kernels are derived are indeed carried out for a period of one year. However, when going back to double check this, we discovered that we had mistook this for the temporal signature and duration of the prescribed atmospheric background state, which for three of the four GCM kernels does extend beyond a single year. We now include a new table (Table 1) that summarizes this and other differences between the GCMs used to derive the GCM kernels and delete the incorrect statement quoted above. If the simulates are for a specific year (which year?), or a climatological run, are they comparable to the CERES-based kernel models which are for the period 2001-2016. No GCM kernel is comparable to the 2001-2016 CERES kernel; background climatologies of ECHAM6, CAM3, and GFDL kernels span several years (or decades) but all pre-date the CERES EBAF era. CAM5's background does fall within the CERES era but is based on a single year only. These discrepancies are why we chose to compare to the mean of all four kernels in our initial performance screening. We chose not to compare the CAM5 kernel to a CERES kernel based on the same background year because the atmospheric state information underlying CAM5 is not based on CERES EBAF (i.e., it would still not be possible to attribute disagreement to differences in the representation of shortwave radiative transfer). This is why we chose instead to emulate CAM5 with the BO18 parameterization run with CAM5's own boundary fluxes. Additionally, I am curious about the inter-annual variability of the multi-year CERES kernels. The interannual variability of a kernel based on CERES can now be inferred from the results of our second application example (Figure 7 C, southern Amazonian deforestation).

3. This study is started with the "need within LULCC science community for simple and transparent tools for predicting radiative forcings from surface albedo changes". Is it possible to provide a simple example of how to apply CACK v1.0 to the LULCC studies? This is a fair request and have thus invested notable effort into demonstrating how both a climatological CACK and a temporally-explicit CACK may be applied to estimate radiative forcings in LULCC studies (New Sections 4 d & e, 5 d & e, and new Figures 6 & 7).

Specific comments:

1. The organization of section 2 and section 3 is a little confusing. The title of section 2 is "Review of existing approaches", but most of the kernels described in section 3 are also "existing approaches", aren't they? We fully agree and have carried out a major re-organization of the manuscript. We are confident that the new manuscript structure is more intuitive and easier to follow and digest.

2. L40, What do you mean by "offline"? Run land surface model offline? Here we mean that GCMs are not practical to apply for estimating albedo change RFs for single locations, and that other modeling approaches have been applied for this purpose involving stand-alone radiative transfer modeling in which the surface and atmosphere are not coupled. I also can't find the paper (Randerson et al. 2006) in the reference. Thank you for pointing out this missing reference which has now been added.

3. L151, Eq. (3) and Eq. (4), are __s and __ the same thing? If yes, it would be better to keep the consistency. Yes, these are the same and have been corrected (thanks).

4. L247, Which part (or period) of data is used for model training, and which part is used for prediction? Model training and prediction datasets are based on a random sampling in both time and space (200,000 grid cells in each). This detail has been added to (new) Section 3 d).

5. L263, It should be "e. Initial screening of candidate models for a CERES-based kernel". Corrected.

6. L409, They are mean absolute bias, not RMSD. Corrected.

7. L441-444, Can the authors explain more about how the land-based solar radiation management is an example of the CACK's flexibility? This was a poorly constructed sentence which has been deleted in the revision.

**Reponses to Anonymous Referee #2**

General comments

The manuscript presented by Bright and O'Halloran suggests the use of a new kernel (CACKv1.0) to derive radiative forcing at the top of the atmosphere from surface albedo changes. This kernel is derived by applying a machine learning technique to identify a formula which can best reproduce the results from kernels derived from Global Circulation Models, once it is applied to CERES satellite-derived data. The authors argue that compared to GCM-derived kernels, this new formula would 1) enable a more transparent derivation of radiative forcing from surface albedo changes, and 2) rely on data from several years. Their analysis shows that the new formula performs better at mimicking the results from GCM-derived kernels compared to previously suggested formulations. They suggest the use of their results by the scientists studying the impacts of land-use and land-cover changes (LULCC) on climate to improve their calculations of radiative forcing from surface albedo changes.

Having an easily applicable kernel that reproduces the results from GCMs can indeed be useful for the LULCC community, and in that sense the authors' initiative is welcome and scientifically significant. Having said that, there are a couple of issues with the authors' approach, while the methodology could be better described to ensure reproducibility of the results. Overall, substantial work also needs to be done on the writing to improve understandability of the manuscript. These issues are not insurmountable, but I recommend that they are addressed before the manuscript is accepted.

We thank Anonymous Referee #2 for his/her constructive feedback. To address his/her major concerns, we have carried out a major re-structuring of the paper that we now believe is easier to follow and more intuitive to digest. This includes more attention to CACK's uncertainties as well as the uncertainties between GCM kernels, and we now include uncertainty estimates for CACK in effort to make CACK v1.0 a more attractive and complete dataset. Lastly, we have also invested notable effort to improve the description of our methods to better-ensure reproducibility of results.

- Specific comments

The real added value of CACK compared to previously suggested simple formulations can only be assessed in light of the uncertainties between GCM kernels. These thus need to be included at least in Figure 1 and discussed in the manuscript, so that the readers can assess for themselves how much of a difference using CACK rather than a simple isotropic kernel (for example) makes. This is a fair comment. We have added additional text describing major sources of uncertainty in GCM-based kernels (new Section 2.a), a new table (new Table 1) highlighting the major differences between them, and a new Figure 1 that now shows the spread among the four GCM kernels we employed (expressed in terms of the seasonal and latitude band mean standard deviations). The authors also mention that the GCM-derived kernels are based on single years of forcing data. This renders them uncertain and thus less appropriate as a benchmark, therefore the authors choose to use the multi-GCM mean kernel as a reference to partly alleviate the lack of consideration of interannual variability when they were derived. This seems reasonable but only partly alleviates the issue. In addition to being explicitly shown and discussed, the uncertainties about GCM-derived kernels (both related to model spread and interannual variability) need to be acknowledged in the Discussion. Even in the current state, more conclusions could be drawn from Figure 1 by describing for example which kernels perform worst against the GCM-derived ones and potentially advancing reasons why this is the case. We believe the revised Figure 1 sufficiently demonstrates the performance of all CERES kernel candidates in light of discrepancies among the GCM kernels themselves.

The methodology should be more detailed to be able to understand how Equation 16 is derived. Which optimal structures and coefficients are considered during the symbolic regression? What should make the reader think that this approach doesn't miss potentially relevant formulas? And which "boundary fluxes (or system parameters derived from these fluxes) that minimized the sum of squared residuals. . ." were considered? This information should at least be provided in the Supplementary Material. This is a fair comment and have thus provided more detail surrounding Eq. (16) (now Eq. (17)) in (new) Section 2.d, including what fluxes were included and what constraints were applied, as well as providing other detail in a new section of the Supporting Information. In the Supporting Information we provide examples of alternate model structures obtained from the machine learning exercise, their performance metrics, and the criteria we applied in the model selection process.

It is also not so clear from the current manuscript why certain choices were made regarding the GCM and kernel selections. Why are four GCM kernels included in the study, are these the only ones available? Correct, these are the only four GCM kernels available at the time the study commenced. We add this rationale to the main text (new Section 2 a). Is there some information existing on the quality of these kernels that guided the selection? Could the authors justify why they "emulated" the kernels of just two GCMs in a second step? Only ECHAM6 and CAM5 kernels were used in the emulation exercise because these were the only two kernels for which the boundary fluxes were also provided (which were needed for the machine learning-based model selection and for kernel emulation). We add this justification to (new) Section 3 b. It seems like only the 3 kernels performing best against the GCM-derived ones were retained for further analysis, but this is also not explicitly mentioned. We have added a sentence at the end of (new) Section 4 a explicitly stating why only these three kernels were retained for further analysis (i.e., they were the top performers of the initial CERES candidate model evaluation exercise).

The structure of the manuscript could be improved to facilitate understandability. For example, why not mentioning the isotropic and anisotropic kernels, as well as the kernel from Qu and Hall in Section 2 already. Currently, at first it may read like they have been derived by the authors. The names of the studies that introduced other types of statistical kernels could also be added in the subsection titles to help the reader follow. We agree that our manuscript needed a more logical organization to facilitate improved readability. We believe the new organization leaves the reader with zero doubt about the origin of the CERES model candidates we consider in the paper.

The description of the CERES dataset also seems misplaced in Section 2. Additionally, in some occurrences the subsection numbering is wrong and the placeholders for Figures or Tables misplaced. We agree and have re-structured the manuscript accordingly such that description of the CERES EBAF v4 products is now provided up front in the Introduction. We have checked and updated all section/table/figure numbering.

Last but not least, the CACK dataset is only mentioned in the conclusion, although from the title it sounds like an important output of the study. If this is the case, it would need to be introduced in the abstract and the introduction of the manuscript. But ultimately, one may wonder whether describing CACK as a dataset is appropriate. Could the authors maybe develop on what makes it more than just applying Eq. 16 to CERES data, for example in terms of pre-processing or perspectives for updates, etc.? We agree that the value of CACK v1.0 packaged as a dataset (i.e., more than just Eq. (17) applied to CERES data) ought to be highlighted and clearly showcased. We have therefore invested considerable effort into describing and quantifying the various sources of uncertainty of CACK and include these as part of a more comprehensive CACK v1.0 data product. We believe this addition strengthens the credibility of CACK v1.0 as a data product and as a viable tool for the advancement of a verification framework for biogeophysical climate forcings on land.

- Technical comments l. 68: "An additional downside is the that". Check typo Corrected typo.
l. 157: to facilitate understandability it could be good to repeat the downsides of GCMderived kernels here We agree and include this as part of (new) Section 2 a.
l. 425: "course" should read "coarse" Corrected.
l. 704-705: can the authors make clearer what is meant by "100X100 sample grid"? Clarified.

[revised manuscript text omitted]

Field Code Changed

Field Code Changed

---

## Referee Report (RR1)

I would like to thank the authors for the revisions they have provided. They clearly put time and effort in addressing the comments and suggestions we made. The manuscript reads a lot better now, and with the information added for example on the uncertainty, it is easier to appreciate the results to the fullest. I still have a list of rather minor comments which I would like to be addressed before publication, but I think this version of the manuscript is close to final.

- l. 27 and after: "CERES-based kernel" reads more appropriate than "CERES kernel", as CACK is derived from CERES data but not included in the same dataset and not produced by the same researchers.

- l.89: can you make the acronym more explicit?

- L.268: "Suggested" or "proposed" is more appropriate than "novel", which may be understood as "recent"

- L. 274: I think I understand what the authors mean, but were the shortwave boundary fluxes really directly compared with the GCM kernel? To me it reads rather like a shortened description of the actual methodological step the authors undertook, in which case I think a more exact description would be necessary. It is still not crystal-clear which CERES variables were considered as potential predictors for the tested models (on l. 271 the authors only refer to "GCM boundary fluxes", which is less restrictive than "shortwave boundary fluxes").

- L. 275: where do the ~200,000, 50%, 97% and 32% numbers come from? How many years were taken into account for each GCM?

- L. 295: I believe it is now Section 2a

- L. 296: consider adding "introduced from Section 3b to 3d" after "six simple model candidates" to clarify the procedure. There could be some confusion with the model candidates examined by the machine learning algorithm.

- L. 321: actually if I understand correctly only two candidate models were used for the emulation

- L. 380: are the subscripts of the numerators of the right-hand term correct? Isn't it supposed to be alpha_CRO,m and alpha_EBF,m? In which case the covariance term between these two variables should also be included, I believe?

- L. 394: "interannal"

- L. 434: "RMSE" instead of "RMSD"

- L. 476: "in all months." Specify that this is on a global average.

- L. 494: There is an interesting pattern on Fig. 5D, where one can observe, in each hemisphere, a thin band located between 40° and 60° where the relative error is

higher. Can the authors advance reasons for this pattern? And do they know what happens over Eastern China?

- L. 498: if I understand well the results from Figure 6. should be understood as "what happens if these pixels were initially completely covered by evergreen broadleaved forests which would then be replaced by grasslands". This is different than "what happens if all evergreen broadleaved forests in these regions were to be replaced by grasslands", but I think the authors should provide more explanation to avoid that these wrong conclusions are drawn by readers.

- L. 508: To be exact, the effect of an increasing albedo trend also emerges, right?

- L. 566: It reads peculiar to have just one subsection.

- L. 8564: the authors could clarify that the "CACK model candidates" are not those of the selection phase by the machine learning algorithm

- L. 896: "mean local" over which domain?

---

## Author Response (AR2)

**Replies to Reviewer #2**

I would like to thank the authors for the revisions they have provided. They clearly put time and effort in addressing the comments and suggestions we made. The manuscript reads a lot better now, and with the information added for example on the uncertainty, it is easier to appreciate the results to the fullest. I still have a list of rather minor comments which I would like to be addressed before publication, but I think this version of the manuscript is close to final.

We thank the reviewer for his/her time to review our revised manuscript and to provide additional constructive feedback.

- l. 27 and after: "CERES-based kernel" reads more appropriate than "CERES kernel", as CACK is derived from CERES data but not included in the same dataset and not produced by the same researchers.

We have changed all references to a "CERES kernel" to a "CERES-based kernel" as per suggested.

- l.89: can you make the acronym more explicit?

Unfortunately, we do not understand this suggestion. To us, CERES albedo change kernel (CACK) is short and descriptive.

- L.268: "Suggested" or "proposed" is more appropriate than "novel", which may be understood as "recent"

We have changed "novel" to "proposed" as per suggested.

- L. 274: I think I understand what the authors mean, but were the shortwave boundary fluxes really directly compared with the GCM kernel? To me it reads rather like a shortened description of the actual methodological step the authors undertook, in which case I think a more exact description would be necessary. It is still not crystal-clear which CERES variables were considered as potential predictors for the tested models (on l. 271 the authors only refer to "GCM boundary fluxes", which is less restrictive than "shortwave boundary fluxes").

We have clarified that only the "shortwave" boundary fluxes were employed in the machine learning exercise and have listed them in the text.

- L. 275: where do the ~200,000, 50%, 97% and 32% numbers come from? How many years were taken into account for each GCM?

Thank you for identifying this confusion. We deleted one stray 50% that was misplaced. Because of the vastly different resolutions of these two models, we chose to maximize the number of (random) pixels included in the lower resolution (ECHAM, 200,000 pixels= 97%), because the same number of pixels was 32% of the other model (CAM5). In both cases, 50% of those pixels were used for training and the other 50% for validation. GCM kernels only represent one year and the input climatologies are indicated in Table 1.

- L. 295: I believe it is now Section 2a

Corrected.

- L. 296: consider adding "introduced from Section 3b to 3d" after "six simple model candidates" to clarify the procedure. There could be some confusion with the model candidates examined by the machine learning algorithm.

We have added the additional explanatory text as per suggested.

- L. 321: actually if I understand correctly only two candidate models were used for the emulation

This is correct, but as this is a result of the initial performance screening (presented subsequently in section 5a), we opted to keep it vague here in section 4b. However, to be consistent with the description of the work flow given at the end of section 1 (Introduction), we have added the word "top" before "candidate models" followed by the text "(as identified from the initial performance screening described in section 4a)" for additional clarity.

- L. 380: are the subscripts of the numerators of the right-hand term correct? Isn't it supposed to be alpha_CRO,m and alpha_EBF,m? In which case the covariance term between these two variables should also be included, I believe?

Yes, the subscripts of the last numerator term in Eq. (22) are correct. We see no reason why the surface albedo of "EBF" and "CRO" should co-vary and have thus followed the uncertainty propagation rules for arithmetic involving two independent variables (as subsequently given as Eq. (23)).

- L. 394: "interannal"

Corrected.

- L. 434: "RMSE" instead of "RMSD"

Corrected.

- L. 476: "in all months." Specify that this is on a global average.

Corrected.

- L. 494: There is an interesting pattern on Fig. 5D, where one can observe, in each hemisphere, a thin band located between 40° and 60° where the relative error is higher. Can the authors advance reasons for this pattern? And do they know what happens over Eastern China?

These appear to be regions with low annual mean clearness indices ("T" of Table 2) and low annual mean incident solar radiation ("SW_down_sfc"), thus giving low annual mean kernel values ("CACK").

- L. 498: if I understand well the results from Figure 6. should be understood as "what happens if these pixels were initially completely covered by evergreen broadleaved forests which would then be replaced by grasslands". This is different than "what happens if all evergreen broadleaved forests in these regions were to be replaced by grasslands", but I think the authors should provide more explanation to avoid that these wrong conclusions are drawn by readers.

This is a fair point. We have added clarifying text to Figure 6's caption and to section 5d making it clear that Figure 6 shows the "local" or "within-grid cell" radiative forcing from a hypothetical land cover conversion.

- L. 508: To be exact, the effect of an increasing albedo trend also emerges, right?

Yes, this is correct. We have clarified.

- l. 520: do the authors already see perspectives for an update of their dataset?

No.

- L. 566: It reads peculiar to have just one subsection.

We agree and have removed "a." before the subsection heading "Concluding Remarks".

- L. 854: the authors could clarify that the "CACK model candidates" are not those of the selection phase by the machine learning algorithm

Fair point. We have clarified this in Figure 1's caption.

- L. 896: "mean local" over which domain?

We delete the term "local" and instead add clarifying text stating that "delta_F is the mean of all grid cells plotted in panel A)."

**Replies to Reviewer #1**

I appreciate the revision. Authors have addressed most of my concerns in the revision. However, I still find a few that were not clarified and I listed them below.

We thank the reviewer for his/her time to review our revised manuscript.

P17, L367-368: "the difference between cropland and evergreen broadleaved forests", but in the caption of Figure 6, it is the difference between grasslands and evergreen broadleaved forests. It is grassland or cropland, or both?

We thank the reviewer for pointing out this inconsistency. We have only simulated the conversion to "croplands" and have revised Figure 6's caption accordingly.

Figure 6: Why is there a clear boundary (like a square) in some regions of South America?

The clear boundaries stem from the albedo product we applied which is based on a multi-scale hierarchical look-up method. Details are provided in Gao et al. (2014), but essentially, if the minimum required number of homogenous land cover albedo/BRDF samples at the target resolution (i.e., n = 80 for the 1° × 1° product) is not found, albedo/BRDF statistics are based on the mean at the next coarsest spatial resolution layer where the minimum required number of samples is found (i.e., either 5° × 5° or global).

Figure 7 B: Is this a trend of annual mean cloud fraction? How do we know if this trend is related to deforestation or climate variability? Is there also decreased cloud cover in other regions without deforestation, which is masked out from the map?

Yes. We do not know if this is related to climate variability or deforestation, but this attribution is beside the point -- we are only concerned in showing the effect of a changing cloud cover on the estimated albedo change radiative forcing. We clarify in section 4e that our demonstration (and hence the result presented in Figure 7) is based on a pixel subset defined by cells where both positive surface albedo and negative cloud area trends occur.

[revised manuscript text omitted]